# From random walks to distances on unweighted graphs

**Tatsunori B. Hashimoto**
MIT EECS
thashim@mit.edu

**Yi Sun**
MIT Mathematics
yisun@mit.edu

**Tommi S. Jaakkola**
MIT EECS
tommi@mit.edu

## Abstract

Large unweighted directed graphs are commonly used to capture relations between entities. A fundamental problem in the analysis of such networks is to properly define the similarity or dissimilarity between any two vertices. Despite the significance of this problem, statistical characterization of the proposed metrics has been limited.

We introduce and develop a class of techniques for analyzing random walks on graphs using stochastic calculus. Using these techniques we generalize results on the degeneracy of hitting times and analyze a metric based on the Laplace transformed hitting time (LTHT). The metric serves as a natural, provably well-behaved alternative to the expected hitting time. We establish a general correspondence between hitting times of the Brownian motion and analogous hitting times on the graph. We show that the LTHT is consistent with respect to the underlying metric of a geometric graph, preserves clustering tendency, and remains robust against random addition of non-geometric edges. Tests on simulated and real-world data show that the LTHT matches theoretical predictions and outperforms alternatives.

## 1 Introduction

Many network metrics have been introduced to measure the similarity between any two vertices. Such metrics can be used for a variety of purposes, including uncovering missing edges or pruning spurious ones. Since the metrics tacitly assume that vertices lie in a latent (metric) space, one could expect that they also recover the underlying metric in some well-defined limit. Surprisingly, there are nearly no known results on this type of consistency. Indeed, it was recently shown [19] that the expected hitting time degenerates and does not measure any notion of distance.

We analyze an improved hitting-time metric – Laplace transformed hitting time (LTHT) – and rigorously evaluate its consistency, cluster-preservation, and robustness under a general network model which encapsulates the latent space assumption. This network model, specified in Section 2, posits that vertices lie in a latent metric space, and edges are drawn between nearby vertices in that space. To analyze the LTHT, we develop two key technical tools. We establish a correspondence between functionals of hitting time for random walks on graphs, on the one hand, and limiting Itô processes (Corollary 4.4) on the other. Moreover, we construct a weighted random walk on the graph whose limit is a Brownian motion (Corollary 4.1). We apply these tools to obtain three main results.

First, our Theorem 3.5 recapitulates and generalizes the result of [19] pertaining to degeneration of expected hitting time in the limit. Our proof is direct and demonstrates the broader applicability of the techniques to general random walk based algorithms. Second, we analyze the Laplace transformed hitting time as a one-parameter family of improved distance estimators based on random walks on the graph. We prove that there exists a scaling limit for the parameter $\beta$ such that the LTHT can become the shortest path distance (Theorem S5.2) or a consistent metric estimator averaging over many paths (Theorem 4.5). Finally, we prove that the LTHT captures the advantages

of random-walk based metrics by respecting the cluster structure (Theorem 4.6) and robustly recovering similarity queries when the majority of edges carry no geometric information (Theorem 4.9). We now discuss the relation of our work to prior work on similarity estimation.

**Quasi-walk metrics:** There is a growing literature on graph metrics that attempts to correct the degeneracy of expected hitting time [19] by interpolating between expected hitting time and shortest path distance. The work closest to ours is the analysis of the phase transition of the $p$-resistance metric in [1] which proves that $p$-resistances are nondegenerate for some $p$; however, their work did not address consistency or bias of $p$-resistances. Other approaches to quasi-walk metrics such as logarithmic-forest [3], distributed routing distances [16], truncated hitting times [12], and randomized shortest paths [8, 21] exist but their statistical properties are unknown. Our paper is the first to prove consistency properties of a quasi-walk metric.

**Nonparametric statistics:** In the nonparametric statistics literature, the behavior of $k$-nearest neighbor and $\varepsilon$-ball graphs has been the focus of extensive study. For undirected graphs, Laplacian-based techniques have yielded consistency for clusters [18] and shortest paths [2] as well as the degeneracy of expected hitting time [19]. Algorithms for exactly embedding $k$-nearest neighbor graphs are similar and generate metric estimates, but require knowledge of the graph construction method, and their consistency properties are unknown [13]. Stochastic differential equation techniques similar to ours were applied to prove Laplacian convergence results in [17], while the process-level convergence was exploited in [6]. Our work advances the techniques of [6] by extracting more robust estimators from process-level information.

**Network analysis:** The task of predicting missing links in a graph, known as link prediction, is one of the most popular uses of similarity estimation. The survey [9] compares several common link prediction methods on synthetic benchmarks. The consistency of some local similarity metrics such as the number of shared neighbors was analyzed under a single generative model for graphs in [11]. Our results extend this analysis to a global, walk-based metric under weaker model assumptions.

## 2 Continuum limits of random walks on networks

### 2.1 Definition of a spatial graph

We take a generative approach to defining similarity between vertices. We suppose that each vertex $i$ of a graph is associated with a latent coordinate $x_i \in \mathbb{R}^d$ and that the probability of finding an edge between two vertices depends solely on their latent coordinates. In this model, given only the unweighted edge connectivity of a graph, we define natural distances between vertices as the distances between the latent coordinates $x_i$. Formally, let $\mathcal{X} = \{x_1, x_2, \ldots\} \subset \mathbb{R}^d$ be an infinite sequence of points drawn i.i.d. from a differentiable density with bounded log gradient $p(x)$ with compact support $D$. A spatial graph is defined by the following:

**Definition 2.1** (Spatial graph). *Let $\varepsilon_n : \mathcal{X}_n \to \mathbb{R}_{>0}$ be a local scale function and $h : \mathbb{R}_{\geq 0} \to [0, 1]$ a piecewise continuous function with $h(x) = 0$ for $x > 1$, $h(1) > 0$, and $h$ left-continuous at $1$. The spatial graph $G_n$ corresponding to $\varepsilon_n$ and $h$ is the random graph with vertex set $\mathcal{X}_n$ and a directed edge from $x_i$ to $x_j$ with probability $p_{ij} = h(|x_i - x_j|\varepsilon_n(x_i)^{-1})$.*

This graph was proposed in [6] as the generalization of $k$-nearest neighbors to isotropic kernels. To make inference tractable, we focus on the large-graph, small-neighborhood limit as $n \to \infty$ and $\varepsilon_n(x) \to 0$. In particular, we will suppose that there exist scaling constants $g_n$ and a deterministic continuous function $\bar{\varepsilon} : D \to \mathbb{R}_{>0}$ so that

$$g_n \to 0, \qquad g_n n^{\frac{1}{d+2}} \log(n)^{-\frac{1}{d+2}} \to \infty, \qquad \varepsilon_n(x)g_n^{-1} \to \bar{\varepsilon}(x) \text{ for } x \in \mathcal{X}_n,$$

where the final convergence is uniform in $x$ and a.s. in the draw of $\mathcal{X}$. The scaling constant $g_n$ represents a bound on the asymptotic sparsity of the graph.

We give a few concrete examples to make the quantities $h$, $g_n$, and $\varepsilon_n$ clear.

1. The directed $k$-nearest neighbor graph is defined by setting $h(x) = 1_{x \in [0,1]}$, the indicator function of the unit interval, $\varepsilon_n(x)$ the distance to the $k^{\text{th}}$ nearest neighbor, and $g_n = (k/n)^{1/d}$ the rate at which $\varepsilon_n(x)$ approaches zero.

2. A Gaussian kernel graph is approximated by setting $h(x) = \exp(-x^2/\sigma^2)1_{x \in [0,1]}$. The truncation of the Gaussian tails at $\sigma$ is an analytic convenience rather than a fundamental limitation, and the bandwidth can be varied by rescaling $\varepsilon_n(x)$.

## 2.2 Continuum limit of the random walk

Our techniques rely on analysis of the limiting behavior of the simple random walk $X_t^n$ on a spatial graph $G_n$, viewed as a discrete-time Markov process with domain $D$. The increment at step $t$ of $X_t^n$ is a jump to a random point in $\mathcal{X}_n$ which lies within the ball of radius $\varepsilon_n(X_t^n)$ around $X_t^n$. We observe three effects: (A) the random walk jumps more frequently towards regions of high density; (B) the random walk moves more quickly whenever $\varepsilon_n(X_t^n)$ is large; (C) for $\varepsilon_n$ small and a large step count $t$, the random variable $X_t^n - X_0^n$ is the sum of many small independent (but not necessarily identically distributed) increments. In the $n \to \infty$ limit, we may identify $X_t^n$ with a continuous-time stochastic process satisfying (A), (B), and (C) via the following result, which is a slight strengthening of [6, Theorem 3.4] obtained by applying [15, Theorem 11.2.3] in place of the original result of Stroock-Varadhan.

**Theorem 2.2.** *The simple random walk $X_t^n$ converges uniformly in Skorokhod space $\mathsf{D}([0,\infty), \overline{D})$ after a time scaling $\widehat{t} = tg_n^2$ to the Itô process $Y_{\widehat{t}}$ valued in the space of continuous functions $\mathsf{C}([0,\infty), \overline{D})$ with reflecting boundary conditions on $D$ defined by*

$$dY_{\widehat{t}} = \nabla \log(p(Y_{\widehat{t}}))\bar{\varepsilon}(Y_{\widehat{t}})^2/3d\widehat{t} + \bar{\varepsilon}(Y_{\widehat{t}})/\sqrt{3}dW_{\widehat{t}}. \tag{1}$$

Effects (A), (B), and (C) may be seen in the stochastic differential equation (1) as follows. The direction of the drift is controlled by $\nabla \log(p(Y_{\widehat{t}}))$, the rate of drift is controlled by $\bar{\varepsilon}(Y_{\widehat{t}})^2$, and the noise is driven by a Brownian motion $W_{\widehat{t}}$ with location-dependent scaling $\bar{\varepsilon}(Y_{\widehat{t}})/\sqrt{3}$.[1]

We view Theorem 2.2 as a method to understand the simple random walk $X_t^n$ through the continuous walk $Y_{\widehat{t}}$. Attributes of stochastic processes such as stationary distribution or hitting time may be defined for both $Y_{\widehat{t}}$ and $X_t^n$, and in many cases Theorem 2.2 implies that an appropriately-rescaled version of the discrete attribute will converge to the continuous one. Because attributes of the continuous process $Y_{\widehat{t}}$ can reveal information about proximity between points, this provides a general framework for inference in spatial graphs. We use hitting times of the continuous process to a domain $E \subset D$ to prove properties of the hitting time of a simple random walk on a graph via the limit arguments of Theorem 2.2.

# 3 Degeneracy of expected hitting times in networks

The hitting time, commute time, and resistance distance are popular measures of distance based upon the random walk which are believed to be robust and capture the cluster structure of the network. However, it was shown in a surprising result in [19] that on undirected geometric graphs the scaled expected hitting time from $x_i$ to $x_j$ converges to inverse of the degree of $x_j$.

In Theorem 3.5, we give an intuitive explanation and generalization of this result by showing that if the random walk on a graph converges to any limiting Itô process in dimension $d \geq 2$, the scaled expected hitting time to any point converges to the inverse of the stationary distribution. This answers the open problem in [19] on the degeneracy of hitting times for directed graphs and graphs with general degree distributions such as directed $k$-nearest neighbor graphs, lattices, and power-law graphs with convergent random walks. Our proof can be understood as first extending the transience or neighborhood recurrence of Brownian motion for $d \geq 2$ to more general Itô processes and then connecting hitting times on graphs to their Itô process equivalents.

## 3.1 Typical hitting times are large

We will prove the following lemma that hitting a given vertex quickly is unlikely. Let $T_{x_j,n}^{x_i}$ be the hitting time to $x_j$ of $X_t^n$ started at $x_i$ and $T_E^{x_i}$ be the continuous equivalent for $Y_{\widehat{t}}$ to hit $E \subset D$ .

**Lemma 3.1** (Typical hitting times are large). *For any $d \geq 2$, $c > 0$, and $\delta > 0$, for large enough $n$ we have $\mathbb{P}(T^{x_i}_{x_j,n} > cg_n^{-2}) > 1 - \delta$.*

To prove Lemma 3.1, we require the following tail bound following from the Feynman-Kac theorem.

**Theorem 3.2** ([10, Exercise 9.12] Feynman-Kac for the Laplace transform). *The Laplace transform of the hitting time (LTHT) $u(x) = \mathbb{E}[\exp(-\beta T^x_E)]$ is the solution to the boundary value problem with boundary condition $u|_{\partial E} = 1$:*

$$\frac{1}{2} Tr[\sigma^T H(u)\sigma] + \mu(x) \cdot \nabla u - \beta u = 0.$$

This will allow us to bound the hitting time to the ball $B(x_j, s)$ of radius $s$ centered at $x_j$.

**Lemma 3.3.** *For $x, y \in D$, $d \geq 2$, and any $\delta > 0$, there exists $s > 0$ such that $\mathbb{E}[e^{-T^x_{B(y,s)}}] < \delta$.*

*Proof.* We compare the Laplace transformed hitting time of the general Itô process to that of Brownian motion via Feynman-Kac and handle the latter case directly. Details are in Section S2.1. □

We now use Lemma 3.3 to prove Lemma 3.1.

*Proof of Lemma 3.1.* Our proof proceeds in two steps. First, we have $T^{x_i}_{x_j,n} \geq T^{x_i}_{B(x_j,s),n}$ a.s. for any $s > 0$ because $x_j \in B(x_j, s)$, so by Theorem 2.2, we have

$$\lim_{n \to \infty} \mathbb{E}[e^{-T^{x_i}_{x_j,n} g_n^{-2}}] \leq \lim_{n \to \infty} \mathbb{E}[e^{-T^{x_i}_{B(x_j,s),n} g_n^{-2}}] = \mathbb{E}[e^{-T^{x_i}_{B(x_j,s)}}]. \tag{2}$$

Applying Lemma 3.3, we have $\mathbb{E}[e^{-T^{x_i}_{B(x_j,s)}}] < \frac{1}{2}\delta e^{-c}$ for some $s > 0$. For large enough $n$, this combined with (2) implies $\mathbb{P}(T^{x_i}_{x_j,n} \leq cg_n^{-2})e^{-c} < \delta e^{-c}$ and hence $\mathbb{P}(T^{x_i}_{x_j,n} \leq cg_n^{-2}) < \delta$. □

### 3.2 Expected hitting times degenerate to the stationary distribution

To translate results from Itô processes to directed graphs, we require a regularity condition. Let $q_t(x_j, x_i)$ denote the probability that $X^n_t = x_j$ conditioned on $X^n_0 = x_i$. We make the following technical conjecture which we assume holds for all spatial graphs.

($\star$) For $t = \Theta(g_n^{-2})$, the rescaled marginal $nq_t(x, x_i)$ is a.s. eventually uniformly equicontinuous.[2]

Let $\pi_{X^n}(x)$ denote the stationary distribution of $X^n_t$. The following was shown in [6, Theorem 2.1] under conditions implied by our condition ($\star$) (Corollary S2.6).

**Theorem 3.4.** *Assuming ($\star$), for $a^{-1} = \int p(x)^2 \bar{\varepsilon}(x)^{-2} dx$, we have the a.s. limit*

$$\hat{\pi}(x) := \lim_{n \to \infty} n\pi_{X^n}(x) = a\frac{p(x)}{\bar{\varepsilon}(x)^2}.$$

We may now express the limit of expected hitting time in terms of this result.

**Theorem 3.5.** *For $d \geq 2$ and any $i, j$, we have*

$$\frac{\mathbb{E}[T^{x_i}_{x_j,n}]}{n} \xrightarrow{a.s.} \frac{1}{\hat{\pi}(x_j)}.$$

*Proof.* We give a sketch. By Lemma 3.1, the random walk started at $x_i$ does not hit $x_j$ within $cg_n^{-2}$ steps with high probability. By Theorem S2.5, the simple random walk $X^n_t$ mixes at exponential rate, implying in Lemma S2.8 that the probability of first hitting at step $t > cg_n^{-2}$ is approximately the stationary distribution at $x_j$. Expected hitting time is then shown to approximate the expectation of a geometric random variable. See Section S2 for a full proof. □

Theorem 3.5 is illustrated in Figures 1A and 1B, which show with only 3000 points, expected hitting times on a $k$-nearest neighbor graph degenerates to the stationary distribution. [3]

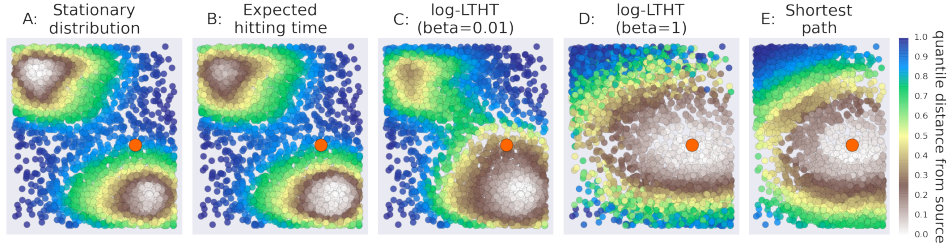

Figure 1: Estimated distance from orange starting point on a $k$-nearest neighbor graph constructed on two clusters. A and B show degeneracy of hitting times (Theorem 3.5). C, D, and E show that log-LTHT interpolate between hitting time and shortest path.

## 4 The Laplace transformed hitting time (LTHT)

In Theorem 3.5 we showed that expected hitting time is degenerate because a simple random walk mixes before hitting its target. To correct this we penalize longer paths. More precisely, consider for $\widehat{\beta} > 0$ and $\beta_n = \widehat{\beta} g_n^2$ the *Laplace transforms* $\mathbb{E}[e^{-\widehat{\beta} T_E^x}]$ and $\mathbb{E}[e^{-\beta_n T_{E,n}^x}]$ of $T_E^x$ and $T_{E,n}^x$.

These Laplace transformed hitting times (LTHT's) have three advantages. First, while the expected hitting time of a Brownian motion to a domain is dominated by long paths, the LTHT is dominated by direct paths. Second, the LTHT for the Itô process can be derived in closed form via the Feynman-Kac theorem, allowing us to make use of techniques from continuous stochastic processes to control the continuum LTHT. Lastly, the LTHT can be computed both by sampling and in closed form as a matrix inversion (Section S3). Now define the scaled log-LTHT as

$$-\log(\mathbb{E}[e^{-\beta_n T_{x_j,n}^{x_i}}])/\sqrt{2\beta_n} g_n.$$

Taking different scalings for $\beta_n$ with $n$ interpolates between expected hitting time ($\beta_n \to 0$ on a fixed graph) and shortest path distance ($\beta_n \to \infty$) (Figures 1C, D, and E). In Theorem 4.5, we show that the intermediate scaling $\beta_n = \Theta(\widehat{\beta} g_n^2)$ yields a consistent distance measure retaining the unique properties of hitting times. Most of our results on the LTHT are novel for any quasi-walk metric.

While considering the Laplace transform of the hitting time is novel to our work, this metric has been used in the literature in an ad-hoc manner in various forms as a similarity metric for collaboration networks [20], hidden subgraph detection [14], and robust shortest path distance [21]. However, these papers only considered the elementary properties of the limits $\beta_n \to 0$ and $\beta_n \to \infty$. Our consistency proof demonstrates the advantage of the stochastic process approach.

### 4.1 Consistency

It was shown previously that for $n$ fixed and $\beta_n \to \infty$, $-\log(\mathbb{E}[-\beta_n T_{x_j,n}^{x_i}])/\beta_n g_n$ converges to shortest path distance from $x_i$ to $x_j$. We investigate more precise behavior in terms of the scaling of $\beta_n$. There are two regimes: if $\beta_n = \omega(\log(g_n^d n))$, then the shortest path dominates and the LTHT converges to shortest path distance (See Theorem S5.2). If $\beta_n = \Theta(\widehat{\beta} g_n^2)$, the graph log-LTHT converges to its continuous equivalent, which for large $\widehat{\beta}$ averages over random walks concentrated around the geodesic. To show consistency for $\beta_n = \Theta(\widehat{\beta} g_n^2)$, we proceed in three steps: (1) we reweight the random walk on the graph so the limiting process is Brownian motion; (2) we show that log-LTHT for Brownian motion recovers latent distance; (3) we show that log-LTHT for the reweighted walk converges to its continuous limit; (4) we conclude that log-LTHT of the reweighted walk recovers latent distance.

**(1) Reweighting the random walk to converge to Brownian motion:** We define weights using the estimators $\widehat{p}$ and $\widehat{\overline{\varepsilon}}$ for $p(x)$ and $\overline{\varepsilon}(x)$ from [6].

---

times to small out neighbors which corrects this problem and derive closed form solutions (Theorem S2.12). This hitting time is non-degenerate but highly biased due to boundary terms (Corollary S2.14).

**Theorem 4.1.** *Let $\widehat{p}$ and $\widehat{\varepsilon}$ be consistent estimators of the density and local scale and $A$ be the adjacency matrix. Then the random walk $\widehat{X}_t^n$ defined below converges to a Brownian motion.*

$$\mathbb{P}(\widehat{X}_{t+1}^n = x_j \mid \widehat{X}_t^n = x_i) = \begin{cases} \frac{A_{i,j}\widehat{p}(x_j)^{-1}}{\sum_k A_{i,k}\widehat{p}(x_k)^{-1}}\widehat{\varepsilon}(x_i)^{-2} & i \neq j \\ 1 - \widehat{\varepsilon}(x_i)^{-2} & i = j \end{cases}$$

*Proof.* Reweighting by $\widehat{p}$ and $\widehat{\varepsilon}$ is designed to cancel the drift and diffusion terms in Theorem 2.2 by ensuring that as $n$ grows large, jumps have means approaching $0$ and variances which are asymptotically equal (but decaying with $n$). See Theorem S4.1. [4] □

**(2) Log-LTHT for a Brownian motion:** Let $W_t$ be a Brownian motion with $W_0 = x_i$, and let $\overline{T}_{B(x_j,s)}^{x_i}$ be the hitting time of $W_t$ to $B(x_j, s)$. We show that log-LTHT converges to distance.

**Lemma 4.2.** *For any $\alpha < 0$, if $\widehat{\beta} = s^\alpha$, as $s \to 0$ we have*

$$-\log(\mathbb{E}[\exp(-\widehat{\beta}\overline{T}_{B(x_j,s)}^{x_i})])/\sqrt{2\widehat{\beta}} \to |x_i - x_j|.$$

*Proof.* We consider hitting time of Brownian motion started at distance $|x_i - x_j|$ from the origin to distance $s$ of the origin, which is controlled by a Bessel process. See Subsection S6.1 for details. □

**(3) Convergence of LTHT for $\beta_n = \Theta(\widehat{\beta}g_n^2)$:** To compare continuous and discrete log-LTHT's, we will first define the $s$-neighborhood of a vertex $x_i$ on $G_n$ as the graph equivalent of the ball $B(x_i, s)$.

**Definition 4.3** ($s$-neighborhood). *Let $\widehat{\varepsilon}(x)$ be the consistent estimate of the local scale from [6] so that $\widehat{\varepsilon}(x) \to \overline{\varepsilon}(x)$ uniformly a.s. as $n \to \infty$. The $\widehat{\varepsilon}$-weight of a path $x_{i_1} \to \cdots \to x_{i_l}$ is the sum $\sum_{m=1}^{l-1} \widehat{\varepsilon}(x_{i_m})$ of vertex weights $\widehat{\varepsilon}(x_i)$. For $s > 0$ and $x \in G_n$, the $s$-neighborhood of $x$ is*

$$\mathsf{NB}_n^s(x) := \{y \mid \text{there is a path } x \to y \text{ of } \widehat{\varepsilon}\text{-weight} \leq g_n^{-1}s\}.$$

For $x_i, x_j \in G_n$, let $\widehat{T}_{B(x_j,s)}^{x_i}$ be the hitting time of the transformed walk on $G_n$ from $x_i$ to $\mathsf{NB}_n^s(x_j)$. We now verify that hitting times to the $s$-neighborhood on graphs and the $s$-radius ball coincide.

**Corollary 4.4.** *For $s > 0$, we have $g_n^2 \widehat{T}_{\mathsf{NB}_n^s(x_j),n}^{x_i} \xrightarrow{d} \overline{T}_{B(x_j,s)}^{x_i}$.*

*Proof.* We verify that the ball and the neighborhood have nearly identical sets of points and apply Theorem 2.2. See Subsection S6.2 for details. □

**(4) Proving consistency of log-LTHT:** Properly accounting for boundary effects, we obtain a consistency result for the log-LTHT for small neighborhood hitting times.

**Theorem 4.5.** *Let $x_i, x_j \in G_n$ be connected by a geodesic not intersecting $\partial D$. For any $\delta > 0$, there exists a choice of $\widehat{\beta}$ and $s > 0$ so that if $\beta_n = \widehat{\beta}g_n^2$, for large $n$ we have with high probability*

$$\left| -\log(\mathbb{E}[\exp(-\beta_n \widehat{T}_{\mathsf{NB}_n^s(x_j),n}^{x_i})])/\sqrt{2\widehat{\beta}} - |x_i - x_j| \right| < \delta.$$

*Proof of Theorem 4.5.* The proof has three steps. First, we convert to the continuous setting via Corollary 4.4. Second, we show the contribution of the boundary is negligible. The conclusion follows from the explicit computation of Lemma S6.1. Full details are in Section S6. □

The stochastic process limit based proof of Theorem 4.5 implies that the log-LTHT is consistent and robust to small perturbations to the graph which preserve the same limit (Supp. Section S8).

## 4.2 Bias

Random walk based metrics are often motivated as recovering a cluster preserving metric. We now show that the log-LTHT of the un-weighted simple random walk preserves the underlying cluster structure. In the 1-D case, we provide a complete characterization.

**Theorem 4.6.** *Suppose the spatial graph has $d = 1$ and $h(x) = 1_{x \in [0,1]}$. Let $T^{x_i}_{\mathsf{NB}^{\hat{\varepsilon}(x_j)g_n}_n(x_j),n}$ be the hitting time of a simple random walk from $x_i$ to the out-neighborhood of $x_j$. It converges to*

$$-\log(\mathbb{E}[-\beta T^{x_i}_{\mathsf{NB}^{\hat{\varepsilon}(x_j)g_n}_n(x_j),n}])/\sqrt{8\beta} \to \int_{x_i}^{x_j} \sqrt{m(x)}dx + o\left(\log(1 + e^{-\sqrt{2\beta}})/\sqrt{2\beta}\right),$$

*where $m(x) = \frac{2}{\bar{\varepsilon}(x)^2} + \frac{1}{\beta}\frac{\partial \log(p(x))}{\partial x^2} + \frac{1}{\beta}\left(\frac{\partial \log(p(x))}{\partial x}\right)^2$ defines a density-sensitive metric.*

*Proof.* Apply the WKBJ approximation for Schrodinger equations to the Feynman-Kac PDE from Theorem 3.2. See Corollary S7.2 and Corollary S2.13 for a full proof. □

The leading order terms of the density-sensitive metric appropriately penalize crossing regions of large changes to the log density; this is not the case for the expected hitting time (Theorem S2.12).

## 4.3 Robustness

While shortest path distance is a consistent measure of the underlying metric, it breaks down catastrophically with the addition of a single non-geometric edge and does not meaningfully rank vertices that share an edge. In contrast, we show that LTHT breaks ties between vertices via the resource allocation (RA) index, a robust local similarity metric under Erdős-Rényi-type noise. [5]

**Definition 4.7.** *The noisy spatial graph $G_n$ over $\mathcal{X}_n$ with noise terms $q_1(n)$, ..., $q_n(n)$ is constructed by drawing an edge from $x_i$ to $x_j$ with probability*

$$p_{ij} = h(|x_i - x_j|\varepsilon_n(x_i)^{-1})(1 - q_j(n)) + q_j(n).$$

Define the directed RA index in terms of the out-neighborhood set $\mathsf{NB}_n(x_i)$ and the in-neighborhood set $\mathsf{NB}^{\mathsf{in}}_n(x_i)$ as $R_{ij} := \sum_{x_k \in \mathsf{NB}_n(x_i) \cap \mathsf{NB}^{\mathsf{in}}_n(x_j)} |\mathsf{NB}_n(x_k)|^{-1}$ and two step log-LTHT by $M^{\mathsf{ts}}_{ij} := -\log(\mathbb{E}[\exp(-\beta T^{x_i}_{x_j,n}) \mid T^{x_i}_{x_j,n} > 1])$. [6] We show two step log-LTHT and RA index give equivalent methods for testing if vertices are within distance $\varepsilon_n(x)$.

**Theorem 4.8.** *If $\beta = \omega(\log(g^d_n n))$ and $x_i$ and $x_j$ have at least one common neighbor, then*

$$M^{\mathsf{ts}}_{ij} - 2\beta \to -\log(R_{ij}) + \log(|\mathsf{NB}_n(x_i)|).$$

*Proof.* Let $P_{ij}(t)$ be the probability of going from $x_i$ to $x_j$ in $t$ steps, and $H_{ij}(t)$ the probability of not hitting before time $t$. Factoring the two-step hitting time yields

$$M^{\mathsf{ts}}_{ij} = 2\beta - \log(P_{ij}(2)) - \log\left(1 + \sum_{t=3}^{\infty} \frac{P_{ij}(t)}{P_{ij}(2)} H_{ij}(t)e^{-\beta(t-2)}\right).$$

Let $k_{\max}$ be the maximal out-degree in $G_n$. The contribution of paths of length greater than 2 vanishes because $H_{ij}(t) \leq 1$ and $P_{ij}(t)/P_{ij}(2) \leq k^2_{\max}$, which is dominated by $e^{-\beta}$ for $\beta = \omega(\log(g^n n))$. Noting that $P_{ij}(2) = \frac{R_{ij}}{|\mathsf{NB}_n(x_i)|}$ concludes. For full details see Theorem S9.1. □

For edge identification within distance $\varepsilon_n(x)$, the RA index is robust even at noise level $q = o(g^{d/2}_n)$.

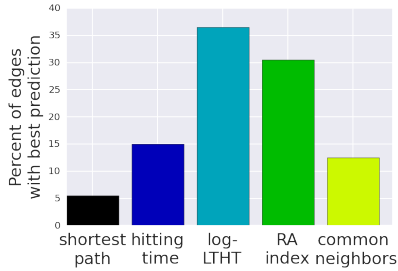

Figure 2: The LTHT recovered deleted edges most consistently on a citation network

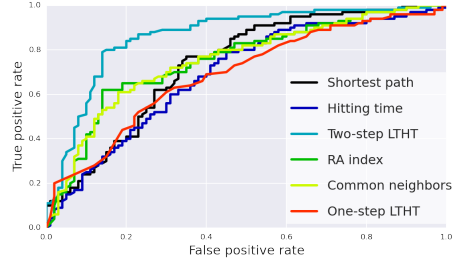

Figure 3: The two-step LTHT (defined above Theorem 4.8) outperforms others at word similarity estimation including the basic log-LTHT.

**Theorem 4.9.** *If $q_i = q = o(g_n^{d/2})$ for all $i$, for any $\delta > 0$ there are $c_1, c_2$ and $h_n$ so that for any $i, j$, with probability at least $1 - \delta$ we have*

- $|x_i - x_j| < \min\{\varepsilon_n(x_i), \varepsilon_n(x_j)\}$ *if $R_{ij}h_n < c_1$;*
- $|x_i - x_j| > 2\max\{\varepsilon_n(x_i), \varepsilon_n(x_j)\}$ *if $R_{ij}h_n > c_2$.*

*Proof.* The minimal RA index follows from standard concentration arguments (see S9.2). □

## 5 Link prediction tasks

We compare the LTHT against other baseline measures of vertex similarity: shortest path distance, expected hitting time, number of common neighbors, and the RA index. A comprehensive evaluation of these quasi-walk metrics was performed in [8] who showed that a metric equivalent to the LTHT performed best. We consider two separate link prediction tasks on the largest connected component of vertices of degree at least five, fixing $\beta = 0.2$.[7] The degree constraint is to ensure that local methods using number of common neighbors such as the resource allocation index do not have an excessive number of ties. Code to generate figures in this paper are contained in the supplement.

**Citation network:** The KDD 2003 challenge dataset [5] includes a directed, unweighted network of e-print arXiv citations whose dense connected component has 11,042 vertices and 222,027 edges. We use the same benchmark method as [9] where we delete a single edge and compare the similarity of the deleted edge against the set of control pair of vertices $i, j$ which do not share an edge. We count the fraction of pairs on which each method rank the deleted edge higher than all other methods. We find that LTHT is consistently best at this task (Figure 2). [8]

**Associative Thesaurus network:** The Edinburgh associative thesaurus [7] is a network with a dense connected component of 7754 vertices and 246,609 edges in which subjects were shown a set of ten words and for each word was asked to respond with the first word to occur to them. Each vertex represents a word and each edge is a weighted, directed edge where the weight from $x_i$ to $x_j$ is the number of subjects who responded with word $x_j$ given word $x_i$.

We measure performance by whether strong associations with more than ten responses can be distinguished from weak ones with only one response. We find that the LTHT performs best and that preventing one-step jumps is critical to performance as predicted by Theorem 4.8 (Figure 3).

## 6 Conclusion

Our work has developed an asymptotic equivalence between hitting times for random walks on graphs and those for diffusion processes. Using this, we have provided a short extension of the proof for the divergence of expected hitting times, and derived a new consistent graph metric that is theoretically principled, computationally tractable, and empirically successful at well-established link prediction benchmarks. These results open the way for the development of other principled quasi-walk metrics that can provably recover underlying latent similarities for spatial graphs.

## Footnotes

[1]Both the variance $\Theta(\varepsilon_n(x)^2)$ and expected value $\Theta(\nabla \log(p(x))\varepsilon_n(x)^2)$ of a single step in the simple random walk are $\Theta(g_n^2)$. The time scaling $\widehat{t} = tg_n^2$ in Theorem 2.2 was chosen so that as $n \to \infty$ there are $g_n^{-2}$ discrete steps taken per unit time, meaning the total drift and variance per unit time tend to a non-trivial limit.

[2]Assumption ($\star$) is related to smoothing properties of the graph Laplacian and is known to hold for undirected graphs [4]. No directed analogue is known, and [6] conjectured a weaker property for all spatial graphs. See Section S1 for further details.

[3]Surprisingly, [19] proved that 1-D hitting times diverge despite convergence of the continuous equivalent. This occurs because the discrete walk can jump past the target point. In Section S2.4, we consider 1-D hitting

[4]This is a special case of a more general theorem for transforming limits of graph random walks (Theorem S4.1). Figure S1 shows that this modification is highly effective in practice.

[5]Modifying the graph by changing fewer than $g^2_n/n$ edges does not affect the continuum limit of the random graph, and therefore preserve the LTHT with parameter $\beta = \Theta(g^2_n)$. While this weak bound allows on average $o(1)$ noise edges per vertex, it does show that the LTHT is substantially more robust than shortest paths without modification. See Section S8 for proofs.

[6]The conditioning $T^{x_i}_{x_j,n} > 1$ is natural in link-prediction tasks where only pairs of disconnected vertices are queried. Empirically, we observe it is critical to performance (Figure 3).

[7]Results are qualitatively identical when varying $\beta$ from 0.1 to 1; see supplement for details.

[8]The two-step LTHT is not shown since it is equivalent to the LTHT in missing link prediction.

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
