[Supplementary Material · supplement.pdf]

# Supplementary materials and proofs

October 30, 2015

## Contents

## 1 Uniform equicontinuity of the marginals

We discuss condition ($\star$) stated in the main text. We conjecture and assume that the following technical condition holds on all spatial graphs.

($\star$) For $t = \Theta(g_n^{-2})$, the rescaled marginal density $nq_t(x, x_i)$ is a.s. eventually uniformly equicontinuous.

To make the terminology we use in ($\star$) clear, we rephrase it as follows.

**Definition S1.1** (Condition $\star$). *If $t = \Theta(g_n^{-2})$, with probability 1, for any $\delta > 0$, there exist $\gamma > 0$ and $n_0$ so that for $n > n_0$, any $x_k \in \mathcal{X}_n$, and any $x_i, x_j \in \mathcal{X}_n$ with $|x_i - x_j| < \gamma$, we have*

$$|nq_t(x_i, x_k) - nq_t(x_j, x_k)| < \delta.$$

This statement allows us to convert the weak convergence in distribution ensured by Theorem 2.2 and the results of (Stroock & Varadhan, 1971) to the convergence in density required by Corollary S2.6. Such a statement rules out the possibility that the density function $q_t(x, x_i)$ oscillates at frequencies increasing with $n$ as $n \to \infty$. Controlling regularity of $q_t(x, x_i)$ seems to be a critical ingredient for approaching $(\star)$.

In the case of an undirected graph, $(\star)$ follows from results in the literature. The strong local limit law for simple random walks shown in (Croydon & Hambly, 2008) yields an even stronger result than $(\star)$. In addition, in the same setting, the convergence result for spectral clustering of (von Luxburg et al., 2008) yields an equicontinuity result for eigenvectors of the Laplacian which implies $(\star)$.

However, for directed graphs no such result exists, and non-reversibility of the Markov chain seems to be an obstacle to proving such a result. Some results on utilizing the directed Laplacian as a smoothing operator (Zhou et al., 2005) exist in this direction. We believe these techniques could lead to an approach to $(\star)$ but thus far they have not yielded a sufficiently strong equicontinuity result.

*Remark* 1. In Hashimoto et al. (2015), a similar conjecture was made on the uniform equicontinuity of the rescaled stationary distribution $n\pi_{X^n}(x)$. We believe "uniform equicontinuity" should be corrected to "eventual uniform equicontinuity" there. In Corollary S2.6, we show that $(\star)$ implies the conjecture of Hashimoto et al. (2015).

One consequence of equicontinuity is that convergence in distribution implies convergence in density. We prove this for the marginal distribution $\widehat{q}_{\widehat{t}}(x_k, x_i)$ of $Y_{\widehat{t}}$ for the purpose of Theorem 3.5, following the original strategy of (Hashimoto et al., 2015).

**Lemma S1.2** (Convergence of marginal densities). *If* $t_n g_n^2 = \widehat{t} = \Theta(1)$, *then under* $(\star)$ *we have*

$$\lim_{n \to \infty} nq_{t_n}(x, x_i) = \frac{\widehat{q}_{\widehat{t}}(x, x_i)}{p(x)},$$

*where the convergence is uniform in $x$ and $x_i$.*

*Proof.* The a.s. weak convergence of processes of Theorem 2.2 (which is uniform in $x_i$) implies by (Ethier & Kurtz, 1986, Theorem 4.9.12) that the empirical marginal distribution

$$d\mu_n = \sum_{j=1}^{n} q_{t_n}(x_j, x_i)\delta_{x_j}$$

converges weakly to the marginal distribution $d\mu = \widehat{q}_{\widehat{t}}(x, x_i)dx$ for $Y_{\widehat{t}}$. For any $x \in \mathcal{X}$ and $\delta > 0$, weak convergence against the test function $1_{B(x,\delta)}$ yields

$$\sum_{y \in \mathcal{X}_n, |y-x|<\delta} q_{t_n}(y, x_i) \to \int_{|y-x|<\delta} \widehat{q}_{\widehat{t}}(y, x_i)dy.$$

By eventual uniform equicontinuity of $nq_t(x, x_i)$, for any $\varepsilon > 0$ there is small enough $\delta > 0$ so that for all $n$ we have

$$\left| \sum_{y \in \mathcal{X}_n, |y-x|<\delta} q_{t_n}(y, x_i) - |\mathcal{X}_n \cap B(x, \delta)|q_{t_n}(x, x_i) \right| \leq n^{-1}|\mathcal{X}_n \cap B(x, \delta)|\varepsilon,$$

which implies that

$$\lim_{n \to \infty} q_{t_n}(x, x_i)p(x)n = \lim_{\delta \to 0} \lim_{n \to \infty} V_d^{-1}\delta^{-d}nq_{t_n}(x, x_i) \int_{|y-x|<\delta} p(y)dy$$

$$= \lim_{\delta \to 0} \lim_{n \to \infty} V_d^{-1}\delta^{-d}|\mathcal{X}_n \cap B(x, \delta)|q_{t_n}(x, x_i) = \lim_{\delta \to 0} V_d^{-1}\delta^{-d} \int_{|y-x|<\delta} \widehat{q}_{\widehat{t}}(y, x_i)dy = \widehat{q}_{\widehat{t}}(x, x_i).$$

We conclude the desired

$$\lim_{n \to \infty} nq_t(x, x_i) = \frac{\widehat{q}_{\widehat{t}}(x, x_i)}{p(x)},$$

where uniformity in $x$ comes from $(\star)$ and uniformity in $x_i$ comes from uniformity of Theorem 2.2. $\qquad\square$

# 2 Hitting times

In this section, we prove Theorem 3.5 to generalize the result of von Luxburg et al. (2014) on degenerate behavior of hitting times via Lemma 3.1. Our proof consists of two parts. First, we show that by Lemma 3.1 we can make the random walk mix before hitting any point. Next, we use this to show that if the chain is sufficiently mixed, then the expected hitting time is degenerate.

## 2.1 Typical hitting times are large

In this subsection, we give a complete proof of Lemma 3.3, reproduced here. Recall that $T_E^{x_i}$ is the hitting time of $Y_t$ from $x_i$ to a domain $E \subset D$. We will require a more general version of the Feynman-Kac theorem.

**Theorem S2.1** ((Øksendal, 2003, Exercise 9.12) Feynman-Kac). *Let $Z_t$ be an Itô process in $\mathbb{R}^d$ defined by*

$$dZ_t = \mu(Z_t)dt + \sigma(Z_t)dB_t.$$

*For a function $V(x)$ and $T_E^x$ the hitting time to a domain $E \subset D$, the function*

$$u(x) = \mathbb{E}\left[e^{-\int_0^{T_E^x} V(Z_s)ds}\right]$$

*is the solution to the boundary value problem*

$$\frac{1}{2}Tr[\sigma^T Hu\sigma] + \mu(x) \cdot \nabla u - V(x)u = 0$$

*with boundary condition $u|_{\partial E} = 1$.*

**Lemma S2.2.** *For $x, y \in D$, $d \geq 2$, and any $\delta > 0$, there exists $s > 0$ such that $\mathbb{E}[e^{-T_{B(y,s)}^x}] < \delta$.*

*Proof.* We use Feynman-Kac to compare $\mathbb{E}[e^{-T_{B(y,s)}^x}]$ for the general process to that of Brownian motion. By Theorem 3.2, $u_s(x) = \mathbb{E}[e^{-T_{B(y,s)}^x}]$ satisfies the boundary value problem

$$\Delta u_s + 2\nabla p(x) \cdot \nabla u_s - 2u_s \bar{\varepsilon}(x)^{-2} = 0$$

with $u_s|_{B(y,s)} \equiv 1$. This is equivalent to

$$\sum_i \left(\partial_i[p(x)^2 \partial_i u_s] - \frac{2}{d}u_s\frac{p(x)^2}{\bar{\varepsilon}(x)^2}\right) = 0.$$

Set $v_s(x) = p(x)u_s(x)$ and change variables to obtain

$$\sum_i \left(\partial_i[p(x)\partial_i v_s - v_s(x)\partial_i p(x)] - \frac{2}{d}v_s\frac{p(x)}{\bar{\varepsilon}(x)^2}\right) = p(x)\Delta v_s - \Delta p(x)v_s - v_s\frac{2p(x)}{\bar{\varepsilon}(x)^2} = 0,$$

which is equivalent to

$$\Delta v_s - \left(\frac{\Delta p(x)}{p(x)} + 2\bar{\varepsilon}(x)^{-2}\right)v_s = 0$$

with boundary condition $v_s|_{\partial B(y,s)} = 1$. Theorem S2.1 for $V(x) = \frac{\Delta p(x)}{p(x)} + 2\bar{\varepsilon}(x)^{-2}$ implies

$$v_s(x) = \mathbb{E}\left[e^{-\int_0^{\overline{T}_{B(y,s)}^x} \frac{\Delta p(B_r)}{p(B_r)} + 2\bar{\varepsilon}(B_r)^{-2}dr}\right]$$

for $B_t$ Brownian motion started at $x$ and $\overline{T}_{B(y,s)}^x$ the hitting time of $B_t$ to $B(y,s)$. For a constant $C$ depending on $p$ and $\bar{\varepsilon}$, we have

$$u_s(x) = \frac{v_s(x)}{p(x)} \leq \mathbb{E}[e^{-C\overline{T}_{B(y,s)}^x}].$$

Applying Lemma S2.3 with this $C$ implies $u_s(x) < \delta e^{-c}$, as needed. $\qquad\square$

**Lemma S2.3.** *For $x, y \in D$, let $B_t$ be a Brownian motion with reflecting boundary condition in $D$ started at $x$ and $T^x_{B(y,s)}$ its hitting time to $B(y,s)$. Then for any sufficiently small $C, c, \delta > 0$, there exists some $s > 0$ so that*

$$\mathbb{E}[e^{-CT^x_{B(y,s)}}] < \delta e^{-c}.$$

*Proof.* Fix $c > 0$ and $\delta > 0$ small enough so that $e^{-c}\delta < 1/10$. If $|x - y| = p$, by Theorem 3 of Byczkowski et al. (2013), the probability density of $T^x_{B(y,s)}$ started at $x$ if there were no outer boundary is bounded by

$$p(t) < C_1 \frac{s^3(p-s)e^{-\frac{(p-s)^2}{2t}}}{pt^{3/2}} \begin{cases} ((t/s^2)^{\frac{d-3}{2}} + (p/s)^{\frac{d-3}{2}})^{-1} & d > 2 \\ \frac{(p/s+t/s^2)^{1/2}(1+\log(p/s))}{(1+\log(1+t/ps))(1+\log(p/s+t/s^2))} & d = 2 \end{cases}$$

for some constant $C_1$.

**Choosing constants:** We claim that we can choose $s$, $p$, and $r$ with $r > p > s > 0$ so that:

(a) $B(y, r)$ is contained entirely in the domain $D$;

(b1) $\frac{r^{2-d} - p^{2-d}}{r^{2-d} - s^{2-d}} > \max\{1/2, \frac{1-e^{-c}\delta}{1 - \frac{1}{2}e^{-c}\delta}\}$ if $d > 2$;

(b2) $\frac{\log r - \log p}{\log r - \log s} > \max\{1/2, \frac{1-e^{-c}\delta}{1 - \frac{1}{2}e^{-c}\delta}\}$ if $d = 2$;

(c1) $C_1 s^d \left( C^{-1} + \int_1^\infty u^{d/2-2} e^{-(p-s)^2 u} du \right) < \frac{1}{4}\delta e^{-c}$ if $d > 2$;

(c2a) $C_1 s^2 \int_1^\infty e^{-Ct}(ps + t)^{1/2} dt < \frac{1}{8}\delta e^{-c}$ if $d = 2$;

(c2b) $C_1 s^2 (ps + 1)^{1/2} \int_0^1 t^{-3/2} e^{-(p-s)^2/2t} dt < \frac{1}{8}\delta e^{-c}$ if $d = 2$.

For $d > 2$, we have that

$$\frac{\Gamma(d/2 - 1)}{(p-s)^{d-2}} = (p-s)^{2-d} \int_0^\infty x^{d/2-2} e^{-x} dx = \int_0^\infty u^{d/2-2} e^{-(p-s)^2 u} du > \int_1^\infty u^{d/2-2} e^{-(p-s)^2 u} du.$$

Then for $p = 2qr$ and $s = qr$, we have that

$$\frac{r^{2-d} - p^{2-d}}{r^{2-d} - s^{2-d}} = \frac{1 - 2^{d-2}q^{d-2}}{1 - q^{d-2}} > 1 - 2^{d-2}q^{d-2}$$

and that

$$s^d \frac{\Gamma(d/2-1)}{(p-s)^{d-2}} < q^2 r^2 \Gamma(d/2-1)$$

Therefore, sending $r \to 0$ and $q \to 0$ gives a choice of $r > p > s$ satisfying (a), (b1), and (c1) as needed.

For $d = 2$, notice that for $t = u^{-1}$, we have

$$\int_0^1 t^{-3/2} e^{-(p-s)^2/2t} dt = \int_1^\infty u^{-1/2} e^{-(p-s)^2 u/2} du.$$

Observe now that

$$(p-s)^{-1}\Gamma(1/2) = (p-s)^{-1} \int_0^\infty t^{-1/2} e^{-t} dt = \int_0^\infty u^{-1/2} e^{-(p-s)^2 u} du > \int_1^\infty u^{-1/2} e^{-(p-s)^2 u} du,$$

whence we conclude that

$$C_1 \frac{s^2 \sqrt{ps+1}}{p-s} \Gamma(1/2) > C_1 s^2 (ps+1)^{1/2} \int_0^1 t^{-3/2} e^{-(p-s)^2/2t} dt.$$

Again choose $p = 2qr$ and $s = qr$, for which we obtain

$$C_1 s^2 (ps+1)^{1/2} \int_0^1 t^{-3/2} e^{-(p-s)^2/2t} dt < C_1 \Gamma(1/2) qr \sqrt{4q^2 r^2 + 1}.$$

Sending $q$ and $r$ to 0 then yields $r > p > s$ satisfying (a), (b2) because $q \to 0$, (c2a) because $s \to 0$ and $(ps+t)^{1/2} < (1+t)^{1/2}$, and (c2b) by the estimate above.

**Bounding the Laplace transform:** Having chosen $r > p > s > 0$ with the desired properties, we have for any $z \in D$ that

$$\mathbb{E}[e^{-CT^z_{B(y,s)}}] \leq \max_{|x-y|=p} \mathbb{E}[e^{-CT^x_{B(y,s)}}].$$

Our strategy will be to bound $\mathbb{E}[e^{-CT^x_{B(y,s)}}]$ for any $x$ with $|x-y| = p$. Fix such an $x$. Let $E$ be the event that the walk hits $B(y,s)$ before $B(y,r)$. By Theorem 3.17 of Mörters & Peres (2010), the probability of $E$ is $\frac{r^{2-d}-p^{2-d}}{r^{2-d}-s^{2-d}}$ if $d > 2$ and $\frac{\log r - \log p}{\log r - \log s}$ if $d = 2$. By our choice of parameters, this probability is at least $\mathbb{P}(E) > \max\{1/2, \frac{1-e^{-c}\delta}{1-\frac{1}{2}e^{-c}\delta}\}$.

Let $\mathbb{E}'[e^{-CT^x_{B(y,s)}}]$ denote the case where there is no outside boundary. For $d > 2$, we have

$$\mathbb{E}'[e^{-CT^x_{B(y,s)}}] < C_1 s^d \int_0^\infty e^{-Ct} t^{-d/2} e^{-(p-s)^2/2t} dt$$

$$< C_1 s^d \left( C^{-1} + \int_0^1 t^{-d/2} e^{-(p-s)^2/2t} dt \right)$$

$$< C_1 s^d \left( C^{-1} + \int_1^\infty u^{d/2-2} e^{-(p-s)^2 u} du \right)$$

$$< \frac{1}{4} \delta e^{-c}$$

by the choice of $s$ and $p$. For $d = 2$, we have

$$\mathbb{E}'[e^{-CT^x_{B(y,s)}}] < C_1 s^2 \int_0^\infty e^{-Ct} t^{-3/2} e^{-(p-s)^2/2t} (ps+t)^{1/2} dt < \frac{1}{4} \delta e^{-c}$$

again by our choice of $s$ and $p$. Conditioning on $E$, we find that

$$\mathbb{E}[e^{-CT^x_{B(y,s)}} | E] \leq \mathbb{P}(E)^{-1} \mathbb{E}'[e^{-CT^x_{B(y,s)}}] < \frac{1}{2} \delta e^{-c}.$$

This implies the desired

$$\mathbb{E}[e^{-CT^x_{B(y,s)}}] \leq \mathbb{P}(E) \mathbb{E}[e^{-CT^x_{B(y,s)}}] + (1 - \mathbb{P}(E)) < \delta e^{-c}. \qquad \square$$

## 2.2 Exponential mixing on spatial graphs

In this subsection, we show that mixing rates are exponential on spatial graphs as assuming $(\star)$.

**Lemma S2.4** (Uniform Doeblin condition). *Assuming $(\star)$, there exist $\alpha > 0$ and $K < \infty$ so that for some $n_0 > 0$ and $\hat{t}_0 > 0$, we have for $n > n_0$ and $\hat{t} > \hat{t}_0$ that*

1. $\min_{x,x_i \in \mathcal{X}_n} q^n_{\lceil \hat{t} g_n^{-2} \rceil}(x, x_i) > \frac{\alpha}{n}$;

2. $\max_{x,x_i \in \mathcal{X}_n} q^n_{\lceil \hat{t} g_n^{-2} \rceil}(x, x_i) < \frac{K}{n}$.

*Proof.* By Lemma S1.2, assuming $(\star)$ we have $n q^n_{\lceil \hat{t} g_n^{-2} \rceil}(x, x_i) \to \hat{q}_{\hat{t}}(x, x_i)/p(x)$, where convergence is uniform in $x$ and $x_i$. Therefore, we may choose $n_0 > 0$ and $\hat{t}_0 > 0$ so that for $n > n_0$ and $\hat{t} > \hat{t}_0$, we have for any $x, x_i \in \mathcal{X}_n$ that

$$\min_{x,x_i \in D} \hat{q}_{\hat{t}}(x, x_i)/p(x) \leq n q^n_{\lceil \hat{t} g_n^{-2} \rceil}(x, x_i) \leq \max_{x,x_i \in D} \hat{q}_{\hat{t}}(x, x_i)/p(x),$$

where the first and last quantities are well-defined by compactness of $D$. Taking

$$\alpha = \frac{1}{2} \min_{x,y \in D} \hat{q}_{\hat{t}}(x, y)/p(x) \qquad \text{and} \qquad K = 2 \max_{x,y \in D} \hat{q}_{\hat{t}}(x, y)/p(x)$$

thus fulfills the desired conditions. $\qquad \square$

**Theorem S2.5.** *Then we may choose $\hat{t}_0, n_0 > 0$ and $C, \beta > 0$ so that for $\hat{t} > \hat{t}_0$, $n > n_0$ and $x_i, x \in \mathcal{X}_n$, we have*

$$|q^n_{\lceil \hat{t} g_n^{-2}\rceil}(x, x_i) - \pi_{X_n}(x)| < C\exp(-\beta\hat{t})\pi_{X_n}(x).$$

*Proof.* By Lemma S2.4, the family of processes $X^n_t$ satisfies the uniform Doeblin condition of (Eloranta, 1990, Section 2.8). The claim follows by the consequences for exponential mixing given in the analogue of (Eloranta, 1990, Theorem 2.7). □

**Corollary S2.6.** *Assuming ($\star$), the rescaled stationary distribution $n\pi_{X^n}(x)$ is a.s. eventually uniformly equicontinuous.*

*Proof.* Choose $\alpha, K, \hat{t}_1, n_1$ by Lemma S2.4 so that for $n > n_1, \hat{t} > \hat{t}_1$, we have

$$\frac{\alpha}{n} < q^n_{\lceil \hat{t} g_n^{-2}\rceil}(x, x_i) < \frac{K}{n}.$$

Choose $C, \beta, \hat{t}_2, n_2$ by Theorem S2.5 so that for $n > n_2, \hat{t} > \hat{t}_2$, we have

$$|q^n_{\lceil \hat{t} g_n^{-2}\rceil}(x, x_i) - \pi_{X_n}(x)| < C\exp(-\beta\hat{t})\pi_{X_n}(x).$$

Thus, for $n > \max\{n_1, n_2\}$ and $\hat{t} > \max\{\hat{t}_1, \hat{t}_2\}$, we have

$$|q^n_{\lceil \hat{t} g_n^{-2}\rceil}(x, x_i) - \pi_{X_n}(x)| < C\exp(-\beta\hat{t})\pi_{X_n}(x) < \frac{CK}{n}\exp(-\beta\hat{t}). \tag{1}$$

Now, for any $\gamma > 0$, choose $n_0 > \max\{n_1, n_2\}$ and $\hat{t}_0 > \max\{\hat{t}_1, \hat{t}_2\}$ large enough and $\delta > 0$ so that

- $\frac{CK}{n_0}\exp(-\beta\hat{t}_0) < \gamma/3$;

- by eventual uniform equicontinuity of $nq^n_{\lceil \hat{t}_0 g_n^{-2}\rceil}(x, x_i)$, for $n > n_0$, if $|x - y| < \delta$, then

$$|nq^n_{\lceil \hat{t}_0 g_n^{-2}\rceil}(x, x_i) - nq^n_{\lceil \hat{t}_0 g_n^{-2}\rceil}(y, x_i)| < \gamma/3.$$

Now, for $n > n_0$ and $|x - y| < \delta$, we find that

$$
\begin{aligned}
|n\pi_{X^n}(x) - n\pi_{X^n}(y)| &\leq |nq^n_{\lceil \hat{t}_0 g_n^{-2}\rceil}(x, x_i) - nq^n_{\lceil \hat{t}_0 g_n^{-2}\rceil}(y, x_i)| + |nq^n_{\lceil \hat{t}_0 g_n^{-2}\rceil}(x, x_i) - n\pi_{X^n}(x)| \\
&\quad + |nq^n_{\lceil \hat{t}_0 g_n^{-2}\rceil}(y, x_i) - n\pi_{X^n}(y)| \\
&< \gamma/3 + \frac{2CK}{n}\exp(-\beta\hat{t}_0) \\
&< \gamma,
\end{aligned}
$$

where we apply (1). This implies that $n\pi_{X^n}(x)$ is eventually uniformly equicontinuous, as needed. □

## 2.3 Expected hitting times degenerate to the stationary distribution

For any $x_j$, let $\pi'_{X^n}$ be the stationary distribution of the simple random walk on the graph $G'_n$ formed from $G_n$ by removing $x_j$ and all edges incident to it.

**Lemma S2.7.** *Assuming ($\star$), the rescaled stationary density $n\pi'_{X^n}(x)$ is a.s. eventually uniformly equicontinuous and satisfies*

$$\lim_{n\to\infty} n\pi'_{X^n}(x) = \hat{\pi}(x).$$

*Proof.* Let $q'_t(x, x_i)$ be the marginal distribution of the simple random walk on the modified graph $G'_n$. Because $G'_n$ is also a spatial graph, by (Hashimoto et al., 2015, Theorem 3.4), the time-rescaled simple random walks on $G'_n$ and $G_n$ converge to the same continuous-time Itô process, and we have under ($\star$) that

$$\lim_{n\to\infty} n\pi'_{X^n}(x_i) = \hat{\pi}(x_i),$$

where the convergence is uniform in $x_i$. □

**Lemma S2.8.** *There exist $t_0 > 0$, $n_0 > 0$, and $C, \beta > 0$ so that for all $\widehat{t} > t_0$ and $n > n_0$ and any integer $t > \widehat{t} g_n^{-2}$, we have*

$$\left| n\mathbb{P}\left( T_{x_j,n}^{x_i} = t \mid T_{x_j,n}^{x_i} \geq t \right) - n \sum_{x \in \mathsf{NB}_n^{\mathsf{in}}(x_j)} \frac{\pi'_{X^n}(x)}{|\mathsf{NB}_n(x)|} \right| < C \exp(-\beta t g_n^2).$$

*Proof.* By Theorem S2.5, we may choose $\widehat{t}_0 > 0$, $n_0 > 0$, $C_1 > 0$, and $\beta > 0$ so that for $\widehat{t} > \widehat{t}_0$ and $n > n_1$, we have

$$|q'_{\lceil \widehat{t} g_n^{-2} \rceil - 1}(x, x_i) - \pi'_{X^n}(x)| < C_1 \exp(-\beta \widehat{t}) \pi'_{X^n}(x_j).$$

We claim that the desired result will hold for $\widehat{t}_0$ and this $n_0$.

By definition, we have

$$\mathbb{P}\left( T_{x_j,n}^{x_i} = t \mid T_{x_j,n}^{x_i} \geq t \right) = \sum_{x \in \mathsf{NB}_n^{\mathsf{in}}(x_j)} \frac{q'_{t-1}(x, x_i)}{|\mathsf{NB}_n(x)|}$$

from which we conclude that for $t > \widehat{t}_0 g_n^{-2}$ we have

$$\left| \mathbb{P}\left( T_{x_j,n}^{x_i} = t \mid T_{x_j,n}^{x_i} \geq t \right) - \sum_{x \in \mathsf{NB}_n^{\mathsf{in}}(x_j)} \frac{\pi'_{X^n}(x)}{|\mathsf{NB}_n(x)|} \right| \leq \left| \sum_{x \in \mathsf{NB}_n^{\mathsf{in}}(x_j)} \frac{q'_{t-1}(x, x_i) - \pi'_{X^n}(x)}{|\mathsf{NB}_n(x)|} \right|$$

$$< C_1 \exp(-\beta t g_n^2) \sum_{x \in \mathsf{NB}_n^{\mathsf{in}}(x_j)} \frac{\pi'_{X^n}(x)}{|\mathsf{NB}_n(x)|}$$

$$< C_1 \exp(-\beta t g_n^2) \frac{|\mathsf{NB}_n^{\mathsf{in}}(x_j)|}{\min_x |\mathsf{NB}_n(x)|} \max_x \pi'_{X^n}(x).$$

We now show there exists $C_2 > 0$ such that $\frac{|\mathsf{NB}_n^{\mathsf{in}}(x_j)|}{\min_x |\mathsf{NB}_n(x)|} < C_2$ almost surely due to the construction of $g_n$ and $\overline{\varepsilon}$. By the out-degree estimate of an isotropic graph (Hashimoto et al., 2015, Theorem S3.2)[1], we have

$$\frac{|\mathsf{NB}_n(x)|}{|\mathcal{X}_n \cap B(x, \varepsilon_n(x))|} \to C(h) p(x)$$

for some constant $C(h)$ independent of $x$ and $n$. Further, since the number of points in $|\mathcal{X}_n \cap B(x, \varepsilon_n(x))| \sim \mathrm{Pois}(\varepsilon_n(x)^d V_d p(x))$, we obtain for a constant $0 < C_x < \infty$ dependent on $p, V_d$ and $C(h)$ that

$$\frac{|\mathsf{NB}_n(x)|}{\varepsilon_n(x)^d n} \to C_x.$$

For the denominator $\min_x |\mathsf{NB}_n(x)|$, the above limit immediately implies that $\min_x |\mathsf{NB}_n(x)| \varepsilon_n^{-d} n^{-1} \to \min_x C_x > 0$ by the lower bounds on $p(x)$ and $\varepsilon_n(x)$.

For the numerator, note that by construction of $\overline{\varepsilon}$, for any $\delta > 0$ there exists a $n$ such that $\varepsilon_n(x)^{-d} < (1 + \delta) \max_x \overline{\varepsilon}(x)^{-d} g_n^{-d}$ almost surely. By the expectation in (Hashimoto et al., 2015, Theorem S3.2), the out-neighborhood of a graph constructed with uniform scale $\max_x \overline{\varepsilon}(x) g_n$ asymptotically dominate the in-neighborhood of the original graph. Therefore,

$$\max_x |\mathsf{NB}_n(x)^{\mathsf{in}}| \overline{\varepsilon}(x)^{-d} g_n^{-d} n^{-1} < \max_x C_x (1 + \delta) < \infty.$$

Combining the two bounds gives that

$$\frac{|\mathsf{NB}_n^{\mathsf{in}}(x_j)|}{\min_x |\mathsf{NB}_n(x)|} < (1 + \delta) \frac{\max_x C_x \overline{\varepsilon}(x)^d}{\min_x C_x \overline{\varepsilon}(x)^d}.$$

The ratio $\frac{\max_x C_x}{\min_x C_x}$ is bounded by definition of $p(x)$, and therefore there exists $C_2 > 0$ such that $\frac{|\mathsf{NB}_n^{\mathsf{in}}(x_j)|}{\min_x |\mathsf{NB}_n(x)|} < C_2$ almost surely. Finally, by Lemma S2.7, there exists $C_3 > 0$ such that $\pi'_{X^n}(x) \leq C_3/n$ for large enough $n$. The original statement follows by setting $C = C_1 C_2 C_3$. $\square$

**Lemma S2.9.** *We have the limit*

$$\lim_{n\to\infty} \sum_{x\in \mathsf{NB}_n^{\mathsf{in}}(x_j)} \frac{1}{|\mathsf{NB}_n(x)|} = 1.$$

*Proof.* We will proceed through three estimates.

**Estimating** $\bar{\varepsilon}(x)$ **for** $x \in \mathsf{NB}_n^{\mathsf{in}}(x_j)$**:** For $\sigma > 0$, define $\gamma = \sigma \min_x \bar{\varepsilon}(x) > 0$. We may choose $\delta > 0$ so that if $|x - y| < \delta$, then $|\bar{\varepsilon}(x) - \bar{\varepsilon}(y)| < \gamma$. Choose $n_0$ so that if $n > n_0$ then $g_n \max_x \bar{\varepsilon}(x) < \delta/2$. For $n > n_0$, we find that for $x \in \mathsf{NB}_n^{\mathsf{in}}(x_j)$, we have

$$|x - x_j| \le \varepsilon_n(x) \le g_n \max_x \bar{\varepsilon}(x) < \delta$$

and therefore that

$$|\bar{\varepsilon}(x) - \bar{\varepsilon}(x_j)| < \sigma \min_x \bar{\varepsilon}(x) \le \sigma\bar{\varepsilon}(x_j).$$

This implies that for $n > n_0$ we have

$$(1 - \sigma)\bar{\varepsilon}(x_j) < \bar{\varepsilon}(x) < (1 + \sigma)\bar{\varepsilon}(x_j). \tag{2}$$

**Estimating** $|\mathsf{NB}_n(x)|$ **for** $x \in \mathsf{NB}_n^{\mathsf{in}}(x_j)$**:** By (Hashimoto et al., 2015, Theorem S3.2), we have

$$\frac{|\mathsf{NB}_n(x)|}{|\mathcal{X}_n \cap B(x, \varepsilon_n(x))|} \to C(h)p(x)$$

for some constant $C(h)$ independent of $x$ and $n$.[2] For any $\tau > 0$, we may therefore find some $n_1$ so that for $n > n_1$ we have

$$(1 - \tau)C(h)p(x)|\mathcal{X}_n \cap B(x, \varepsilon_n(x))| < |\mathsf{NB}_n(x)| < (1 + \tau)C(h)p(x)|\mathcal{X}_n \cap B(x, \varepsilon_n(x))|.$$

On the other hand, by (2), for $x \in \mathsf{NB}_n^{\mathsf{in}}(x_j)$ and any $\sigma > 0$ there is some $n_0$ so that for $n > n_0$ we have

$$|\mathcal{X}_n \cap B(x, (1 - \sigma)\varepsilon_n(x_j))| < |\mathcal{X}_n \cap B(x, \varepsilon_n(x))| < |\mathcal{X}_n \cap B(x, (1 + \sigma)\varepsilon_n(x_j))|. \tag{3}$$

**Estimating** $|\mathsf{NB}_n^{\mathsf{in}}(x_j)|$**:** By (2) and an analogue of the proof of (Hashimoto et al., 2015, Theorem S3.2), we have for $x \in \mathsf{NB}_n^{\mathsf{in}}(x_j)$ that for any $\rho > 0$, there is $n_2 > 0$ so that if $n > n_2$ then

$$(1 - \rho)C(h)p(x)|\mathcal{X}_n \cap B(x, (1 - \sigma)\varepsilon_n(x_j))| < |\mathsf{NB}_n^{\mathsf{in}}(x_j)| < (1 + \rho)C(h)p(x)|\mathcal{X}_n \cap B(x, (1 + \sigma)\varepsilon_n(x_j))|. \tag{4}$$

**Completing the proof:** The conclusion follows by taking $\tau, \sigma, \rho \to 0$, choosing $n$ large, and combining (3) and (4). □

**Lemma S2.10.** *The quantity* $\theta_n(x_j) = \sum_{x\in \mathsf{NB}_n^{\mathsf{in}}(x_j)} \frac{\pi'_{X^n}(x)}{|\mathsf{NB}_n(x)|}$ *satisfies*

$$\lim_{n\to\infty} n\theta_n(x_j) = \widehat{\pi}(x_j).$$

*Proof.* Fix a sequence of points $y_1, y_2, \ldots$ in $\mathcal{X}$ with $y_k \in G'_k$ so that $\lim_{k\to\infty} y_k = x_j$. Fix any $\delta > 0$. By Lemma S2.7, we may find some $n_0$ so that for $n > n_0$, for each $x \in \mathsf{NB}_n^{\mathsf{in}}(x_j)$ we have

$$|\pi'_{X^n}(x) - \pi'_{X^n}(y_n)| < \delta/2.$$

This implies that for $n > n_0$ we have

$$\left| n\theta_n(x_j) - n\pi'_{X^n}(y_n) \sum_{x\in \mathsf{NB}_n^{\mathsf{in}}(x_j)} \frac{1}{|\mathsf{NB}_n(x)|} \right| < \frac{\delta}{2} \sum_{x\in \mathsf{NB}_n^{\mathsf{in}}(x_j)} \frac{1}{|\mathsf{NB}_n(x)|}.$$

The result then follows by Lemma S2.9 and Lemma S2.7. □

**Theorem S2.11.** *For any $x_i$ and $x_j$, we have*

$$\frac{\mathbb{E}[T^{x_i}_{x_j,n}]}{n} \to \frac{1}{\widehat{\pi}(x_j)},$$

*where the convergence is a.s. in the draw of $\mathcal{X}$.*

*Proof.* By definition, we have

$$\mathbb{E}[T^{x_i}_{x_j,n} \mid T^{x_i}_{x_j,n} > \widehat{t}g_n^{-2}] \geq \mathbb{E}[T^{x_i}_{x_j,n}] \geq \mathbb{P}(T^{x_i}_{x_j,n} > \widehat{t}g_n^{-2})\mathbb{E}[T^{x_i}_{x_j,n} \mid T^{x_i}_{x_j,n} > \widehat{t}g_n^{-2}]. \tag{5}$$

By Lemma 3.1, for any $\delta > 0$ and $\widehat{t}_0 > 0$, there is some $n_1$ so that for $n > n_1$ and $\widehat{t} > \widehat{t}_0$ we have $\mathbb{P}(T^{x_i}_{x_j,n} > \widehat{t}g_n^{-2}) > (1-\delta)$. Define now $p_t = \mathbb{P}(T^{x_i}_{x_j,n} = t \mid T^{x_i}_{x_j,n} \geq t)$; by definition we have

$$\mathbb{E}[T^{x_i}_{x_j,n} \mid T^{x_i}_{x_j,n} > \widehat{t}g_n^{-2}] = \sum_{t=\lceil \widehat{t}g_n^{-2}\rceil}^{\infty} tp_t \prod_{r=\lceil \widehat{t}g_n^{-2}\rceil}^{t-1} (1-p_r).$$

By Lemma S2.8, we have for some $n_2$ that for $n > n_2$ and $t > \widehat{t}_0 g_n^{-2}$ that

$$|p_t - \theta_n(x_j)| < \frac{C\exp(-\beta t g_n^2)}{n}$$

so in particular for $\delta = \frac{1}{2}\min_{x\in D}\widehat{\pi}(x)$ and $\tau = 2\max_{x\in D}\widehat{\pi}(x)$, we have for some $n_3$ that for $n > n_3$ we have

$$\delta < np_t < \tau \text{ and } \delta < n\theta_n(x_j) < \tau.$$

For $n_4$ large enough that $1 - \tau/n_4 > \delta/n_4$, for $n > n_4$ we have

$$\left| p_t \prod_{r=\lceil \widehat{t}g_n^{-2}\rceil}^{t-1} (1-p_r) - \theta_n(x_j)(1-\theta_n(x_j))^{t-\lceil \widehat{t}g_n^{-2}\rceil}\right| < \sum_{r=\lceil \widehat{t}g_n^{-2}\rceil}^{t-1} \frac{C\exp(-\beta r g_n^2)}{n}(1-\tau/n)^{t-\lceil \widehat{t}g_n^{-2}\rceil-1}$$

$$< \frac{C}{n}\frac{e^{-\beta\widehat{t}}}{1-e^{-\beta g_n^2}}(1-\tau/n)^{t-\lceil \widehat{t}g_n^{-2}\rceil-1}.$$

This implies that

$$\frac{1}{n}\left| \mathbb{E}[T^{x_i}_{x_j,n} \mid T^{x_i}_{x_j,n} > \widehat{t}g_n^{-2}] - \sum_{t=\lceil \widehat{t}g_n^{-2}\rceil}^{\infty} t\theta_n(x_j)(1-\theta_n(x_j))^{t-\lceil \widehat{t}g_n^{-2}\rceil}\right| < \sum_{t=\lceil \widehat{t}g_n^{-2}\rceil}^{\infty} \frac{C}{n^2}\frac{e^{-\beta\widehat{t}}}{1-e^{-\beta g_n^2}}(1-\tau/n)^{t-\lceil \widehat{t}g_n^{-2}\rceil-1}$$

$$< \frac{C}{\tau(n-\tau)}\frac{e^{-\beta\widehat{t}}}{1-e^{-\beta g_n^2}},$$

where we note that for $n > 2\tau$, we have

$$\frac{C}{\tau(n-\tau)}\frac{e^{-\beta\widehat{t}}}{1-e^{-\beta g_n^2}} < \frac{2Ce^{-\beta\widehat{t}}}{\tau}n^{-1}(g_n^{-2} + \frac{1}{2} + \frac{1}{12}g_n^2).$$

Because $\lim_{n\to\infty} n^{-1}(g_n^{-2} + \frac{1}{2} + \frac{1}{12}g_n^2) = 0$, considering $n > \max\{n_1, n_2, n_3, n_4\}$, we conclude that

$$\lim_{n\to\infty}\frac{1}{n}\mathbb{E}[T^{x_i}_{x_j,n} \mid T^{x_i}_{x_j,n} > \widehat{t}g_n^{-2}] = \lim_{n\to\infty}\frac{1}{n}\sum_{t=\lceil \widehat{t}g_n^{-2}\rceil}^{\infty} t\theta_n(x_j)(1-\theta_n(x_j))^{t-\lceil \widehat{t}g_n^{-2}\rceil}$$

$$= \lim_{n\to\infty}\frac{1}{n}\frac{1-\theta_n(x_j)+\theta_n(x_j)\lceil \widehat{t}g_n^{-2}\rceil}{\theta_n(x_j)}$$

$$= \lim_{n\to\infty}\frac{1}{n\theta_n(x_j)} = \frac{1}{\widehat{\pi}(x_j)},$$

where the last equality follows from Lemma S2.10. Now by (5), we conclude that

$$\lim_{n\to\infty}\frac{1}{n}\mathbb{E}[T^{x_i}_{x_j,n}] = \frac{1}{\widehat{\pi}(x_j)}. \qquad \square$$

## 2.4 The case of one dimension

The Laplacian-based bounds in (von Luxburg et al., 2014) suggest that the hitting time should diverge even when the dimension of the underlying geometric graph is 1. This is a very surprising result, since the continuous random walk in one dimension converges to a non-trivial limit. We provide another explanation of this result in our framework.

Intuitively, this happens since we are concerned with the hitting time to a single point, and the discrete random walk may jump over the point, while the continuous walk cannot. To demonstrate this, we show that considering the hitting time to a sufficiently large out-neighborhood of a vertex instead of the vertex itself fixes this problem.

Pick $x_i, x_j \in G_n$, and let $X_t^n$ be the simple random walk on $G_n$. Suppose without loss of generality that $x_i < x_j$ and define

$$\gamma = \inf_n \min_{x_i \in \mathcal{X}_n} x_i$$

to be the left boundary of $D$. Pick a sequence of sets of vertices $S_n \subset \mathcal{X}_n$ so that every element in $S_n$ is reachable from $x_j$ in $o(g_n^{-1})$ steps and the removal of $S_n$ from $G_n$ disconnects $G_n$. Let $T_{S_n}^{x_i}$ be the hitting time to any point in $S_n$. We will use the Feynman-Kac theorem for functionals of hitting time.

**Theorem S2.12** ((Øksendal, 2003, Exercise 9.12) Feynman-Kac). *Let $Z_t$ be an Itô process in $\mathbb{R}^d$ defined by*

$$dZ_t = \mu(Z_t)dt + \sigma(Z_t)dB_t.$$

*For a function $f(x)$ and $T_E^x$ the hitting time to a domain $E \subset D$, the function*

$$u(x) = \mathbb{E}\left[\int_0^{T_E^x} f(Z_s)ds\right]$$

*is the solution to the boundary value problem*

$$\frac{1}{2}Tr[\sigma^T Hu\sigma] + \mu(x) \cdot \nabla u + f(x) = 0$$

*with boundary condition $u|_{\partial E} = 1$.*

**Theorem S2.13.** *Such a sequence of vertex sets $S_n$ always exists and the expected hitting time $\mathbb{E}[T_{S_n,n}^{x_i}]$ converges to a non-degenerate continuum limit defined by*

$$\mathbb{E}[T_{S_n,n}^{x_i} g_n^2] \to \int_{x_i}^{x_j} \frac{1}{p(y)^2} \int_\gamma^y \frac{2p(z)^2}{\overline{\varepsilon}(z)^2} dzdy.$$

*Proof.* First we prove a sequence $S_n$ exists. Take the set of points $\widehat{S}_n = \{x_k : |x_k - x_j| < c_n\}$ for a sequence $c_n$ with $c_n \to 0$ and $c_n g_n \to \infty$. Let $s$ be the maximum shortest path distance to any element in $\widehat{S}_n$. Then we have $s = o(g_n^{-1})$ since $c_n \to 0$ and the length of the shortest path between any two points scales as $\Theta(g_n^{-1})$. Therefore the set $S_n$ defined by all points whose shortest path distance to $x_j$ is at most $s$ fulfills the requirements.

Let $\widehat{T}_{ij}$ be the hitting time to $x_j$ of $Y_{\widehat{t}}$ started at $x_i$. Note that it is not infinite because we have $d = 1$. By Corollary 4.4 and the fact that $sg_n^{-1} \to 0$, we have

$$T_{S_n,n}^{x_i} g_n^2 \xrightarrow{d} \widehat{T}_{ij}.$$

Finally, by Theorem S2.12 with $f(x) \equiv 1$, the expected hitting time $u(x)$ to $x_j$ under the continuous process $Y_{\widehat{t}}$ started at $x$ is the solution to the boundary value problem

$$\frac{1}{2}\overline{\varepsilon}(x)^2 u''(x) + \frac{p'(x)}{p(x)}\overline{\varepsilon}(x)^2 u'(x) + 1 = 0$$

We may rewrite this as

$$p(x)^2 u''(x) + 2p(x)p'(x)u'(x) = -\frac{2p(x)^2}{\overline{\varepsilon}(x)^2},$$

after which integration of both sides and application of $u'(\gamma) = 0$ implies that

$$p(x)^2 u'(x) = -\int_\gamma^x \frac{2p(z)^2}{\overline{\varepsilon}(z)^2} dz.$$

Another integration and application of $u(x_j) = 0$ implies that

$$u(x) = -\int_{x_j}^x \frac{1}{p(y)^2} \int_\gamma^y \frac{2p(z)^2}{\overline{\varepsilon}(z)^2} dz dy.$$

Setting $x = x_i$ then implies that

$$\lim_{n\to\infty} \mathbb{E}[T_{S_n,n}^{x_i} g_n^2] = \mathbb{E}[\widehat{T}_{ij}] = \int_{x_i}^{x_j} \frac{1}{p(y)^2} \int_\gamma^y \frac{2p(z)^2}{\overline{\varepsilon}(z)^2} dz dy. \qquad \square$$

For cases where the kernel function takes values in $\{0, 1\}$, such as the $k$-nearest neighbor graph, the following corollary is useful.

**Corollary S2.14.** *Suppose that $G_n$ is constructed by the kernel $h(x) = 1_{[0,1]}$. Then the expected hitting time of $X_t^n$ started at $x_i$ to the out-neighbors of $x_j$ converges to the limit of Theorem S2.13*

*Proof.* From the fact that the out-neighborhood of $x_j$ satisfies the conditions for $S_n$ in Theorem S2.13.  $\square$

Although this metric is nontrivial in the sense that it retains some information about the latent space metric, it is still highly distorted. We examine this phenomenon in the case of $\overline{\varepsilon}(x) = 1$ and $p(x) = 1$ in the following Corollary.

**Corollary S2.15.** *If $\overline{\varepsilon}(x) = 1$ and $p(x) = 1$ in Corollary S2.14, for any $x_i$ and $x_j$ the rescaled expectation of the hitting time $T_{\mathsf{NB}_n(x_j),n}^{x_i}$ of $X_t^n$ started at $x_i$ to the out-neighborhood of $x_j$ has the limit*

$$\mathbb{E}[T_{\mathsf{NB}_n(x_j),n}^{x_i} g_n^2] \to |x_j - x_i| \cdot |x_j + x_i - 2\gamma|.$$

*Proof.* This follows by applying Theorem S2.13 with our $\overline{\varepsilon}(x)$ and $p(x)$.  $\square$

*Remark* 2. Note that the boundary condition in Corollary 2.15 induces a large non-uniform multiplicative error. Because of this, the expected hitting time is not consistent even in the ideal situation of a one-dimensional latent space with random walk converging to Brownian motion. Compare this result with Theorem 4.5, which shows a much stronger consistency property.

# 3   Computing the LTHT

Algorithmically, computing the LTHT can be done in two major ways: matrix inversion, or sampling. For the results in the paper, we use the direct sampling method of drawing a simple random walk and calculating the exponentially discounted hitting time. This same computation can be performed using a truncated power method (Yazdani, 2013, Algorithm 1).

Alternative approaches for computing the LTHT involve the following matrix inversion method. Let $P$ be the transition matrix for some random walk. Then the LTHT $\mathbb{E}[\exp(-\beta T_{x_j,n}^{x_i})]$ is given by

$$\mathbb{E}[\exp(-\beta T_{x_j,n}^{x_i})] = (I - W \exp(-\beta))_{ji}^{-1}.$$

Note that this expression is a close discrete analog of Feynman-Kac (Theorem 2.1). This relationship was used in prior work (Smith et al., 2014, Eq. 22) to calculate the LTHT in a different setting and formulation. Correctness of this expression can be seen via the series expansion which was computed as a normalizer for randomized shortest paths (Françoisse et al., 2013, Algorithm 2). This method has been used to calculate the LTHT in in prior work (Kivimäki et al., 2014).

# 4 Reweighting the random walk

Recall that $A_{ij}^n$ was the adjacency matrix of $G_n$. In Corollary S4.2, we give a complete proof of Theorem 4.1 from the maintext.

## 4.1 General construction and application to Brownian motion

Let $a_n(x)$ and $b_n(x)$ be scalar functions on $\mathcal{X}_n$ with possibly stochastic dependence on $\mathcal{X}_n$ so that

$$\lim_{n \to \infty} a_n(x) = \overline{a}(x) \qquad \text{and} \qquad \lim_{n \to \infty} b_n(x) = \overline{b}(x)$$

uniformly in $x$ for some deterministic $\overline{a}(x)$ and $\overline{b}(x)$.

**Theorem S4.1.** *If $a_n(x)$ is a.s. eventually equicontinuous, $\overline{a}(x)$ is smooth with bounded gradient, and $\overline{b}(x)$ is continuous and bounded in $(0, 1]$, the weighted random walk $Z_t$ defined by the transition matrix*

$$\mathbb{P}(Z_{t+1} = x_j \mid Z_t = x_i) = \begin{cases} A_{i,j}^n \frac{a_n(x_j)}{\sum_{x_k \in \mathsf{NB}_n(x_i)} a_n(x_k)} b_n(x_i) & i \neq j \\ 1 - b_n(x_i) & i = j \end{cases}$$

*converges to the Itô process with drift $\nabla \log(p(x)\overline{a}(x))/3$ and diffusion $\overline{\varepsilon}(x)^2 \overline{b}(x)/3$.*

*Proof.* To show convergence to an Itô process, it suffices to check the Stroock-Varadhan criterion (Stroock & Varadhan, 1971). Since the boundary for both the original and modified walk are the same, we only need check that

$$\mathbb{E}[Z_{t+1} - x_i \mid Z_t = x_i] \xrightarrow{p} \frac{1}{3} \frac{\nabla[p(x_i)\overline{a}(x_i)]}{p(x_i)\overline{a}(x_i)} \overline{\varepsilon}(x_i)^2 \overline{b}(x_i), \text{ and}$$

$$\mathbb{E}[(Z_{t+1} - x_i)^2 \mid Z_t = x_i] \xrightarrow{p} \frac{1}{3} \overline{\varepsilon}(x_i)^2 \overline{b}(x_i).$$

For this, by definition we have that

$$\mathbb{E}[Z_{t+1} - x_i \mid Z_t = x_i] = \mathbb{P}(Z_{t+1} \neq x_i) \frac{1}{\sum_{x_k \in \mathsf{NB}_n(x_i)} a_n(x_k)} \sum_{x_k \in \mathsf{NB}_n(x_i)} (x_k - x_i) a_n(x_k), \text{ and}$$

$$\mathbb{E}[(Z_{t+1} - x_i)^2 \mid Z_t = x_i] = \mathbb{P}(Z_{t+1} \neq x_i) \frac{1}{\sum_{x_k \in \mathsf{NB}_n(x_i)} a_n(x_k)} \sum_{x_k \in \mathsf{NB}_n(x_i)} (x_k - x_i)^2 a_n(x_k),$$

from which the desired estimates follow by using $\mathbb{P}(Z_{t+1} \neq x_i) = b_n(x_i)$ and the values and concentration of conditional moments $\mathbb{E}[f(Z_t - Z_{t-1}) \mid Z_{t-1}, Z_t \neq Z_{t-1}]$ given by applying Lemma S4.3 and Lemma S4.4 to $f(x) = x$ and $f(x) = x^2$. $\qquad \square$

**Corollary S4.2.** *Let $\widehat{p}$ and $\widehat{\varepsilon}$ be consistent estimators of the density and local scale and $A$ be the adjacency matrix. Then the random walk $\widehat{X}_t^n$ defined by the following transition*

$$\mathbb{P}(\widehat{X}_{t+1}^n = x_j \mid \widehat{X}_t^n = x_i) = \begin{cases} \frac{A_{i,j} \widehat{p}(x_j)^{-1}}{\sum_k A_{i,k} \widehat{p}(x_k)^{-1}} \widehat{\varepsilon}(x_i)^{-2} & i \neq j \\ 1 - \widehat{\varepsilon}(x_i)^{-2} & i = j \end{cases}$$

*converges to a Brownian motion.*

*Proof.* Set $a_n(x) = \widehat{p}(x)^{-1}$ and $b_n(x) = \widehat{\varepsilon}(x)^{-2}$ as estimated by (Hashimoto et al., 2015) so that $\lim_{n \to \infty} a_n(x) = p(x)^{-1}$ and $\lim_{n \to \infty} b_n(x) = \overline{\varepsilon}(x)^{-2}$. These satisfy the conditions of Theorem S4.1 and yield limiting drift and diffusion coefficients for Brownian motion. $\qquad \square$

## 4.2 Technical moment estimates

In this subsection, we give the moment estimates necessary in the proof of Theorem S4.1. We first derive the expected values of each moment quantity averaged over draws of $\mathcal{X}_n$.

**Lemma S4.3** (Expected values of reweighting). *Let $x = X_t^n$ and $y = X_{t+1}^n$. Then the conditional expectation after weighting by $a_n(x)$ converges to the weighted draw over $p(x)a_n(x)$; that is, we have a.s. that*

$$
\lim_{n\to\infty} \left| \frac{1}{h_n} |\mathsf{NB}_n(x)| \, \mathbb{E}\left[ \frac{a_n(y)}{\sum_{z\in\mathsf{NB}_n(x)} a_n(z)} f(y-x) \mid y \neq x \right] \right.
$$
$$
\left. - \frac{1}{h_n} \int_{y\in B(x,\varepsilon_n(x))} f(y-x) \frac{p(y)\overline{a}(y)}{\int_{z\in B(x,\varepsilon_n(x))} p(z)\overline{a}(z)dz} dy \right| = 0.
$$

*Proof.* By the continuity of $p$ and a.s. eventual equicontinuity of $a_n(y)$, we have $\sup_{y\in B(x,\varepsilon_n(x))} |a_n(y)p(y) - a_n(x)p(x)| \to 0$ and $\sup_{y\in B(x,\varepsilon_n(x))} |p(y) - p(x)| \to 0$. These together imply

$$
\frac{\int_{y\in B(x,\varepsilon_n(x))} a_n(y)p(y)dy}{\int_{y\in B(x,\varepsilon_n(x))} p(y)dy} \xrightarrow{a.s.} \overline{a}(x). \tag{6}
$$

Because $a_n(x) \to \overline{a}(x)$ uniformly in $x$, for any $\delta > 0$, we may choose $n_0$ so that for $n > n_0$, we have $|a_n(x) - \overline{a}(x)| < \delta/2$ and $\varepsilon_n(x)$ is small enough so that if $|y - x| < \varepsilon_n(x)$, then $|\overline{a}(y) - \overline{a}(x)| < \delta/2$. For $n > n_0$, we then have

$$
\sup_{z\in\mathsf{NB}_n(x)} |a_n(z) - \overline{a}(x)| \leq \sup_{z\in\mathsf{NB}_n(x)} |a_n(z) - \overline{a}(z)| + |\overline{a}(z) - \overline{a}(x)| < \delta.
$$

This shows that $\sup_{z\in\mathsf{NB}_n(x)} |a_n(z) - \overline{a}(x)| \to 0$ and therefore

$$
\frac{\sum_{z\in\mathsf{NB}_n(x)} a_n(z)}{|\mathsf{NB}_n(x)|} \xrightarrow{a.s.} \overline{a}(x). \tag{7}
$$

Applying (6) and (7), we find that

$$
\lim_{n\to\infty} \left| \frac{1}{h_n} |\mathsf{NB}_n(x)| \, \mathbb{E}\left[ \frac{a_n(y)}{\sum_{z\in\mathsf{NB}_n(x)} a_n(z)} f(y-x) \mid x \neq y \right] \right.
$$
$$
\left. - \frac{1}{h_n} \mathbb{E}\left[a_n(y)f(y-x) \mid x \neq y\right] \frac{\int_{y\in B(x,\varepsilon_n(x))} p(y)dy}{\int_{y\in B(x,\varepsilon_n(x))} a_n(y)p(y)dy} \right| \to 0.
$$

We apply the argument of (Hashimoto et al., 2015, Lemma 3.2) to this iterated expectation to obtain

$$
\lim_{n\to\infty} \left| \frac{1}{h_n} \mathbb{E}\left[a_n(y)f(y-x) \mid x \neq y\right] \frac{\int_{y\in B(x,\varepsilon_n(x))} p(y)dy}{\int_{y\in B(x,\varepsilon_n(x))} a_n(y)p(y)dy} \right.
$$
$$
\left. - \frac{1}{h_n} \int_{y\in B(x,\varepsilon_n(x))} f(y-x) \frac{p(y)\overline{a}(y)}{\int_{z\in B(x,\varepsilon_n(x))} p(z)\overline{a}(z)dz} dy \right| \to 0. \quad \square
$$

Evaluating the integrals for $f(x) = x$ and $f(x) = x^2$ in Lemma S4.3 implies that the expected value of an increment of the reweighted walk across all draws of $\mathcal{X}_n$ limits to $\nabla \log[p(x)\overline{a}(x)]/3$ and the expected variance of the increment limits to $\overline{\varepsilon}(x)^2\overline{b}(x)/3$. However, in order to apply the Stroock-Varadhan criteria we require that this hold with high probability over all draws of $\mathcal{X}_n$.

**Lemma S4.4** (Strong LLN for local moments). *For a function $f(x)$ such that $\sup_{x\in B(0,\varepsilon)} |f(x)| < \varepsilon$ for small $\varepsilon > 0$, we have a.s. that*

$$
\lim_{n\to\infty} \left| \frac{1}{h_n} \sum_{y\in\mathsf{NB}_n(x)} \frac{a_n(y)}{\sum_{z\in\mathsf{NB}_n(x)} a_n(z)} f(y-x) - \frac{1}{h_n} \int_{y\in B(x,\varepsilon_n(x))} f(y-x) \frac{p(y)\overline{a}(y)}{\int_{z\in B(x,\varepsilon_n(x))} p(z)\overline{a}(z)dz} dy \right| = 0.
$$

Figure 1: Distance estimates for various values of $\beta$ on re-weighted walks on a simulated dataset

(a) Simple random walk is biased toward region with high density

(b) Re-weighted walk diffuses evenly on the true metric

Figure 2: Visualization of the marginal distribution $P_{ij}(t)$ of a random walk over a $k$-nn graph on a Gaussian restricted to a disk, starting at the blue initial point and run for 40 steps. The re-weighted walk diffuses evenly from the starting point, ignoring biases due to density $p$ and neighborhood size $\varepsilon$.

*Proof.* Define the quantity

$$\mu(x) = \frac{1}{h_n} \int_{y \in B(x, \varepsilon_n(x))} f(y - x) \frac{p(y)\overline{a}(y)}{\int_{z \in B(x, \varepsilon_n(x))} p(z)\overline{a}(z)dz} dy.$$

We wish to bound

$$p_n(t) = \mathbb{P}\left( \left| \frac{1}{h_n} \sum_{y \in \mathsf{NB}_n(x)} \frac{a_n(y)}{\sum_{z \in \mathsf{NB}_n(x)} a_n(z)} f(y - x) - \mu(x) \right| \geq t \right). \tag{8}$$

By a.s. eventual equicontinuity of $a_n(y)$, we have for some $c > 0$ and large enough $n$ that

$$\frac{a_n(y)}{\sum_{z \in \mathsf{NB}_n(x)} a_n(z)} \leq c \frac{1}{|\mathsf{NB}_n(x)|}.$$

By the construction of $\varepsilon_n(x)$, if $|y - x| < \varepsilon_n(x)$, then $|f(y - x)| \leq \varepsilon_n(x)$. Combining these two we apply Hoeffding's inequality to obtain that

$$p_n(t) \leq 2 \exp\left( -\frac{2h_n^2 |\mathsf{NB}_n(x)|^2 c^2 t^2}{|\mathsf{NB}_n(x)|\varepsilon_n(x)^2} \right) = o(n^{-2t^2\omega(1)}), \tag{9}$$

where we use that $|\mathsf{NB}_n(x)| = \omega\left(n^{2/(d+2)} \log(n)^{d/(d+2)}\right)$. This completes the proof by Borel-Cantelli. $\qquad\square$

Figure 3: Distance estimates for various values of $\beta$ on re-weighted walks on a simulated dataset

# 5    Consistency at $\beta = \omega(\log(g_n^d n))$ via shortest paths

**Definition S5.1.** *Define the f-length of any path $\gamma \subset D$ as given in (Alamgir & von Luxburg, 2012) as*

$$D_{f,\gamma} = \int_\gamma f(\gamma(t))|\gamma'(t)|dt.$$

*Let the f-distance from $x$ to $y$ be the minimum path length between two points*

$$D_f(x,y) = \min_{\gamma \in C^1, \gamma(0)=x, \gamma(1)=y} D_{f,\gamma}.$$

**Theorem S5.2.** *Let $\beta = \omega(\log(g_n^d n))$, then for $f(x) = \bar{\varepsilon}(x)^{-1}$ we have*

$$-\log(\mathbb{E}[\exp(-\beta T_{x_j,n}^{x_i})])/\beta g_n \to D_f(x_i, x_j).$$

*Proof.* Define $H_{ij}(t)$ to be the probability of not hitting $x_j$ by step $t$, and $P_{ij}(t)$ to be the probability of going from $x_i$ to $x_j$ in exactly $t$ steps. The expected value is the series

$$-\log(\mathbb{E}[\exp(-\beta T_{x_j,n}^{x_i})])/\beta g_n = -\beta^{-1} g_n \log\left(\sum_{t=0}^{\infty} P_{ij}(t) H_{ij}(t) \exp(-\beta t)\right).$$

Now, let $D_{ij}$ be the length of the shortest path from $i$ to $j$. By definition $H_{ij}(D_{ij}) = 1$ and

$$-\log(\mathbb{E}[\exp(-\beta T_{x_j,n}^{x_i})])/\beta g_n = D_{ij} g_n - \log(P_{ij}(D_{ij}))\frac{g_n}{\beta} - \log\left(1 + \sum_{t=D_{ij}+1}^{\infty} \frac{P_{ij}(t)}{P_{ij}(D_{ij})} H_{ij}(t) \exp(-\beta(t - D_{ij}))\right)\frac{g_n}{\beta}.$$

This forms the upper bound

$$-\log(\mathbb{E}[\exp(-\beta T_{x_j,n}^{x_i})])/\beta g_n \leq D_{ij} g_n - \log(P_{ij}(D_{ij}))\frac{g_n}{\beta}.$$

The probability $P_{ij}(D_{ij})$ of hitting $x_j$ in exactly $D_{ij}$ steps is lower bounded by $(g_n^d n)^{-D_{ij}}$ since by definition at least one path exists. This implies that $\log(P_{ij}(D_{ij})) = o(g_n^{-1}\log(g_n^d n))$ and therefore

$$D_{ij}g_n \leq -\log(\mathbb{E}[\exp(-\beta T_{x_j,n}^{x_i})])/\beta g_n \leq D_{ij}g_n - o(1),$$

where the lower bound follows because it is impossible to reach vertex $x_j$ in less than $D_{ij}$ steps. By (Alamgir & von Luxburg, 2012) for the $k$-nearest neighbor case and (Hashimoto et al., 2015) with Lemma S6.3 for the other cases of a spatial graph, $D_{ij}g_n$ converges to the $f$-distance defined by $\bar{\varepsilon}(x)^{-1}$, completing the proof. $\square$

# 6 Consistency of LTHT

In this section, we prove some results needed in the proof of Theorem 4.5.

## 6.1 LTHT of the Brownian motion

**Lemma S6.1.** *Let $W_t$ be a Brownian motion with $W_0 = x_i$. Let $\overline{T}_{B(x_j,s)}^{x_i}$ be the hitting time of $W_t$ to $B(x_j, s)$. For any $\alpha < 0$, if $\widehat{\beta} = s^\alpha$, as $s \to 0$ we have*

$$-\log(\mathbb{E}[\exp(-\widehat{\beta}\overline{T}_{B(x_j,s)}^{x_i})])/\sqrt{2\widehat{\beta}} \to |x_i - x_j|.$$

*Proof of Lemma 6.1.* Let $B_t = |W_t|$ be the order $\nu = d/2 - 1$ Bessel process. The LTHT of $B_t$ to hit $x_j \pm s$ is equivalent to the LTHT of $W_t$ to hit $B(x_j, s)$. Defining $w = |x_i - x_j|$, by (Borodin & Salminen, 2002, Eq 4.2.0.1), this is:

$$\mathbb{E}[\exp(-\widehat{\beta}\overline{T}_{B(x_j,s)}^{x_i})] = \frac{K(\nu, w\sqrt{2\widehat{\beta}})w^{-\nu}}{K(\nu, s\sqrt{2\widehat{\beta}})s^{-\nu}},$$

where $K(\nu, w)$ is a modified Bessel function of the second kind. Write $-\log(\mathbb{E}[\exp(-\widehat{\beta}\overline{T}_{B(x_j,s)}^{x_i})])/\sqrt{2\widehat{\beta}} = c_1 + c_2$ for

$$c_1 = -\log(K(\nu, w\sqrt{2\widehat{\beta}})w^{-\nu})/\sqrt{2\widehat{\beta}}$$
$$c_2 = -\log(K(\nu, s\sqrt{2\widehat{\beta}})s^{-\nu})/\sqrt{2\widehat{\beta}}.$$

Taylor expansion of $c_1$ at $\widehat{\beta}^{-1} = 0$ yields

$$c_1 = w - \frac{\log(\pi^2/(8\widehat{\beta})) + 4\log(w^{-1/2-\nu})}{4\sqrt{2\widehat{\beta}}} + o\left(\frac{\nu^2}{w\widehat{\beta}}\right),$$

hence $c_1 \to w$. For $c_2$, note that $\nu\log(s)/\sqrt{2\widehat{\beta}} \to 0$ and for $s$ small,

$$K(\nu, s\sqrt{2\widehat{\beta}}) \sim \begin{cases} -\log(s\sqrt{2\widehat{\beta}}) & d = 2 \\ \frac{1}{2}\Gamma(s\sqrt{2\widehat{\beta}})(\frac{1}{2}s\sqrt{2\widehat{\beta}})^{-\nu} & d > 2 \end{cases}.$$

by (Abramowitz & Stegun, 1972, p375). Checking that $-\log(K(\nu, s\sqrt{2\widehat{\beta}}))/\sqrt{2\widehat{\beta}} \to 0$ and combining estimates gives $-\log(\mathbb{E}[\exp(-\widehat{\beta}\overline{T}_{B(x_j,s)}^{x_i})])/\sqrt{2\widehat{\beta}} = c_1 + c_2 \to w.$ $\square$

## 6.2 Proof of Corollary 4.4

We prove here Corollary S6.2. We recall the setup. For points $x_i, x_j \in G_n$ and $s > 0$, $\widehat{T}^{x_i}_{B(x_j,s)}$ is the hitting time of the de-biased walk on $G_n$ from $x_i$ to $\mathsf{NB}^s_n(x_j)$. In the continuous setting, $\overline{T}^{x_i}_{B(x_j,s)}$ is the hitting time of Brownian motion with reflecting boundary conditions in $D$ from $x_i$ to $B(x_j, s)$. We would like to show the following.

**Corollary S6.2.** *For $s > 0$, we have $g_n^2 \widehat{T}^{x_i}_{B(x_j,s)} \xrightarrow{d} \overline{T}^{x_i}_{B(x_j,s)}$.*

Our proof consists of two steps. First, we show that hitting $\mathsf{NB}^s_n(x_j)$ is equivalent to hitting $B(x_j, s)$ with the discrete walk. Second, we use S4.2 to show convergence in distribution of this second hitting time. We require a few lemmas.

**Lemma S6.3.** *For any $\delta > 0$ and $s > 0$ so that $B(x_j, s + \delta) \subset D$, we have with high probability that*

$$\mathcal{X}_n \cap B(x_j, s - \delta) \subset \mathsf{NB}^s_n(x_j) \subset B(x_j, s + \delta).$$

*Proof.* Recall that $\mathsf{NB}^s_n(x_j)$ is defined as

$$\mathsf{NB}^s_n(x) := \{y \mid \text{there is a path } x \to y \text{ of } \widehat{\varepsilon}\text{-weight} \le s\}.$$

The estimator $\widehat{\varepsilon}(x)$ is appropriately scaled such that $\widehat{\varepsilon}(x) \to \overline{\varepsilon}(x)$ uniformly and almost surely. Thus, we need to show that $\widehat{\varepsilon}$-weighted shortest path distance converges to true shortest path distance up to error $\Theta(g_n)$.

We first present the simpler case of a constant kernel $h(x) \equiv 1$ over $[0, 1]$; this includes the $k$-nearest neighbor and $\varepsilon$-ball cases. Let $D_{ij}$ be the minimum $\widehat{\varepsilon}$-weight of a path from $x_i$ to $x_j$. The proof of (Hashimoto et al., 2015, Theorem S4.5) shows that in this case

$$\left| |x_i - x_j| - D_{ij} g_n \right| \le \varepsilon_n(x_j). \tag{10}$$

If $x_k \in \mathcal{X}_n \cap B(x_j, s - \delta)$, this implies that $D_{jk} g_n \to |x_k - x_j| \le s - \delta$. Therefore $D_{jk} g_n \le s$ with high probability and $x_k \in \mathsf{NB}^s_n(x_j)$. If $x_k \in \mathsf{NB}^s_n(x_j)$, this implies that $D_{ik} \le s$. By Equation (6.2), we have $s \ge D_{ij} g_n \to |x_i - x_j|$. Therefore $x_k \in B(x_j, s + \delta)$ with high probability.

The proof for the case of generic $h(x)$ is closely analogous. The same proof as used for (Hashimoto et al., 2015, Theorem S4.5) shows that there exists some $k$ such that $|x_k - x_j| \le \varepsilon_n(x_k)$ such that

$$\left| |x_i - x_j| - D_{ik} g_n \right| \le \varepsilon_n(x).$$

At this stage, a difference arises. The proof of (Hashimoto et al., 2015, Theorem S4.5) bounds the number of steps necessary to reach distance $\varepsilon_n(x_j)$ to the target, but for a general choice of $h(x)$ this does not guarantee that we can reach $x_j$.

For general $h(x)$, we instead show that two extra jumps are sufficient. Because $h(1) > 0$ and $h$ is continuous at 1, there exists some interval $(c_1, 1)$ and some $c_2 > 0$ such that

$$\inf_{x \in (c_1, 1)} h(x) > c_2.$$

This annulus will yield a lower bound on the true connectivity. If $|x_i - x_j| \le \varepsilon_n(x_i)$, then the probability that there is some point $x_k$ such that the path $x_i \to x_k \to x_j$ exists in $G_n$ is governed by

$$\mathbb{P}(D_{ij} > 2) = (1 - c_2)^{2|\mathsf{NB}_n(x_i) \cap \mathsf{NB}^{in}_n(x_j)|}$$

where

$$|\mathsf{NB}_n(x_i) \cap \mathsf{NB}^{in}_n(x_j)| \sim \mathrm{Pois}(g_n^d n \tau(x_i - x_j))$$

and $\tau(z)$ is the total overlapping density between the connectivity kernel of $x_i$ and $x_j$. This is lower bounded by the annulus; for any $d > 2$ the annuli have nonzero overlap volume and

$$\tau(z) \ge c_2^2 \int_{x \in B(0,1)} 1_{1 > |x| > c_1} 1_{1 > |1 - x| > c_1} dx \ge 0.$$

This implies that $|\mathsf{NB}_n(x_i) \cap \mathsf{NB}_n^{in}(x_j)| = \Theta(k)$ with high probability and therefore

$$\mathbb{P}(D_{ij} > 2) = (1 - c_2)^{2|\mathsf{NB}_n(x_i) \cap \mathsf{NB}_n^{in}(x_j)|} \to 0.$$

Thus, there exists a two step path from $x_i$ to $x_j$ whenever $|x_i - x_j| < \varepsilon_n(x_i)$. Combined with the analogue of (Hashimoto et al., 2015, Theorem S4.5), this shows that with high probability there is a walk of $\widehat{\varepsilon}$-weight at most $|x_i - x_k| + 2\varepsilon_n(x_k)$ from $x_i$ to $x_k$. We conclude in the same way as in the constant kernel case. $\square$

We now require a lemma on the continuity of functions on Skorokhod space. For this, we recall the metric which induces the relevant topology on Skorokhod space. Let $\Lambda$ be the set of strictly increasing continuous bijections $[0, \infty) \to [0, \infty)$. The Skorokhod metric on $\mathsf{D}([0, \infty), \overline{D})$ and $\mathsf{D}([0, \infty), \mathbb{R}_{\geq 0})$ is given by

$$\sigma(f, g) = \inf_{\lambda \in \Lambda} \max\{||\lambda - \mathrm{id}||, ||f - g \circ \lambda||\},$$

where $|| \cdot ||$ denotes the sup-norm on the relevant space.

**Lemma S6.4.** *Let $B \subset D$ be any ball and $\overline{T}_B^x$ the hitting time from $x$ to $B$ of Brownian motion with reflecting boundary condition in $D$. As a map $\mathsf{D}([0, \infty), \overline{D}) \to \mathbb{R}_{\geq 0}$, the hitting time $\overline{T}_B^x$ is continuous on the subset of $\mathsf{C}([0, \infty), \overline{D})$ of paths whose hitting time to $B$ is finite.*

*Proof.* Denote by $\mathsf{C}_B$ the subset of $\mathsf{C}([0, \infty), \overline{D})$ of paths whose hitting time to $B$ is finite. We first claim that the function

$$d_B : \mathsf{D}([0, \infty), \overline{D}) \to \mathsf{D}([0, \infty), \mathbb{R}_{\geq 0})$$

given by composition with the function $\overline{d}_B : \overline{D} \to B$ giving the distance to $B$ is continuous. For any $\varepsilon > 0$, pick $\delta$ by uniform continuity of $\overline{d}_B$ so that $\delta < \varepsilon$ and if $|x - y| < \delta$, then $|\overline{d}_B(x) - \overline{d}_B(y)| < \varepsilon$. If $\sigma(f, g) < \delta$, we have

$$\sigma(d_B(f), d_B(g)) = \inf_{\lambda \in \Lambda} \max\{||\lambda - \mathrm{id}||, ||\overline{d}_B \circ f - \overline{d}_B \circ g \circ \lambda||\}.$$

Because $\sigma(f, g) < \delta$, we may find $\lambda \in \Lambda$ so that $||f - g \circ \lambda|| < \delta$ and $||\lambda - \mathrm{id}|| < \delta$. By our choice of $\delta$, this implies that

$$\max\{||\lambda - \mathrm{id}||, ||\overline{d}_B \circ f - \overline{d}_B \circ g \circ \lambda||\} < \varepsilon$$

and therefore that $\sigma(d_B(f), d_B(g)) < \varepsilon$, establishing continuity.

Now, the image of $\mathsf{C}_B$ under $d_B$ is the subset $\mathsf{C}_0$ of $\mathsf{C}([0, \infty), \mathbb{R}_{\geq 0})$ of paths whose hitting time to 0 is finite. By (Whitt, 1980, Theorem 7.1), the first passage time to 0 is continuous on $\mathsf{C}_0$. The hitting time $\overline{T}_B^x$ is the composition of the first passage time and $d_B$, hence is continuous on $\mathsf{C}_B$ as claimed. $\square$

**Lemma S6.5.** *Let $B \subset D$ be any ball containing at least one point of $G_n$. For $x_i \in G_n$, let $T_{B,n}^{x_i}$ be the hitting time from $x_i$ to $B$ of the de-biased random walk on $G$. Then $g_n^2 \widehat{T}_{B,n}^{x_i} \xrightarrow{d} \overline{T}_B^x$.*

*Proof.* First, note that both the de-biased random walk and Brownian motion with reflecting boundary condition started at $x_i$ have a.s. finite hitting time to $B$. By Lemma S6.4, the hitting time to $B$ is a.s. continuous on the subset of $\mathsf{D}([0, \infty), \overline{D})$ containing their trajectories. The desired convergence in distribution then follows from Corollary S4.2, the continuous mapping theorem (see (Whitt, 1980, Section 1)), and noting the time-rescaling used in Corollary S4.2. $\square$

*Proof of Corollary S6.2.* Recall that $T_{B(x_j, p), n}^{x_i}$ is the hitting time of the simple random walk on $G_n$ to $B(x_j, p)$. By Lemma S6.3, for any $\delta > 0$, we have with high probability that

$$T_{B(x_j, s+\delta), n}^{x_i} \leq \widehat{T}_{B(x_j, s), n}^{x_i} \leq T_{B(x_j, s-\delta), n}^{x_i}.$$

Applying Lemma S6.5 to $B(x_j, s \pm \delta)$, we see that

$$g_n^2 T_{B(x_j, s\pm\delta), n}^{x_i} \xrightarrow{d} \overline{T}_{B(x_j, s\pm\delta)}^{x_i},$$

which shows that

$$\overline{T}_{B(x_j, s-\delta)}^{x_i} \leq \lim_{n \to \infty} g_n^2 \widehat{T}_{B(x_j, s), n}^{x_i} \leq \overline{T}_{B(x_j, s+\delta)}^{x_i}$$

for all $\delta > 0$. Sending $\delta \to 0$ yields the result. $\square$

## 6.3    Proof of Theorem 4.5

We prove here Theorem 6.6. Recall we chose an estimator $\widehat{\varepsilon}(x) \to \overline{\varepsilon}(x)$.

**Theorem S6.6.** *Let $x_i$ and $x_j$ be points in $G_n$ connected by a geodesic not intersecting $\partial D$. For any $\delta > 0$, there exists a choice of $\widehat{\beta}$ and $s > 0$ so that if $\beta = \widehat{\beta}g_n^2$, we have for large $n$ with high probability that*

$$\left| -\log(\mathbb{E}[\exp(-\beta \widehat{T}^{x_i}_{B(x_j,s),n})])/\sqrt{2\widehat{\beta}} - |x_i - x_j| \right| < \delta.$$

Our proof will proceed by converting to the continuous setting by Corollary 6.2 and then reducing to the case of Brownian motion without boundary which was analyzed in Lemma 6.1. Because we are in the setting of Brownian motion with reflecting boundary conditions, we must apply the "principle of not feeling the boundary" to show that our results are unaffected by it. For this, we define some events to condition on.

Let $\mathcal{G}$ be the geodesic from $x_i$ to $x_j$, and for a distance scale $\rho$, let $\mathcal{G}(\rho)$ be the set of all points of distance less than $\rho$ from $\mathcal{G}$. Choose $\rho$ small enough so that $\mathcal{G}(\rho) \subset D$. For a distance $s > 0$, let $B_t$ be a Brownian motion without boundary started at $x_i$, and let $\overline{T}^{x_i}_{B(x_j,s)}$ be its hitting time to $B(x_j,s)$. For a time $t^\star > 0$, define the following events:

- let $E_1$ be the event that $\overline{T}^{x_i}_{B(x_j,s)} < t^\star$;

- let $E_2$ be the event that $E_1$ holds and $B_t$ hits $B(x_j,s)$ before $\mathcal{G}(\rho)$;

- let $E_3$ and $E_4$ denote the analogous events for Brownian motion with boundary.

Notice that $\mathbb{P}(E_2) = \mathbb{P}(E_4)$. In the rest of this section, we will consider the scalings $t^\star = s^\gamma$ and $\widehat{\beta} = s^\alpha$ for some $\gamma > 0$ and $\alpha < 0$ so that $\alpha + \gamma > 0$, so that $\widehat{\beta}t^\star \to \infty$ as $s \to 0$.

Let $p_t^R(x,y)$, $p_t^K(x,y)$, $p_t^G(x,y)$, and $p_t^F(x,y)$ be the transition density of Brownian motion started at $x$ and run for time $t$ with reflecting boundary condition, killed at $\partial D$, killed at $\partial \mathcal{G}(\rho)$, and no boundary condition, respectively. For $\star \in \{R, K, G, F\}$, let $h^\star(T)$ be the probability that the respective process hits $B(x_j,s)$ before time $T$, and let $h^\star(t,x)$ be the density of hitting at $x \in B(x_j,s)$ at time $t$. Note that $p_t^K(x,y) \leq p_t^R(x,y)$, $p_t^K(x,y) \leq p_t^F(x,y)$, and $p_t^G(x,y) \leq p_t^F(x,y)$. We have the following three lemmas, which are instances of "the principle of not feeling the boundary."

**Lemma S6.7.** *For $x,y$ a distance at least $\rho' > 0$ to $\partial \mathcal{G}(\rho)$, there are constants $t_0 > 0$ and $\lambda > 0$ dependent only on $\rho$ so that for $t < t_0$, we have*

$$\frac{p_t^G(x,y)}{p_t^F(x,y)} \geq 1 - e^{-\lambda t^{-1}}.$$

*Proof.* This follows from (Hsu, 1995, Theorem 1.2) and the results of (Varadhan, 1967). ☐

**Lemma S6.8.** *For $x,y$ a distance at least $\rho' > 0$ to $\partial D$, there are constants $t_0 > 0$ and $\lambda > 0$ dependent only on $\rho$ so that for $t < t_0$, we have*

$$\frac{p_t^K(x,y)}{p_t^F(x,y)} \geq 1 - e^{-\lambda t^{-1}}.$$

*Proof.* This follows from (Hsu, 1995, Theorem 1.2) and the results of (Varadhan, 1967). ☐

**Lemma S6.9.** *For $x,y$ a distance at least $\rho' > 0$ to $\partial D$, there are constants $t_0 > 0$ and $\lambda > 0$ dependent only on $\rho$ so that for $t < t_0$, we have*

$$\frac{p_t^K(x,y)}{p_t^R(x,y)} \geq 1 - e^{-\lambda t^{-1}}.$$

*Proof.* Note that our domain $D$ is a Lipschitz domain in the sense of (Bass & Hsu, 1991, Section 3). Therefore, by (Bass & Hsu, 1991, Theorem 3.1, Theorem 3.4, and Remark 3.11), the reflecting Brownian motion in $D$ has transition density $p_t^R(x,y)$ satisfying

$$C_1 t^{-d/2} e^{-c_1 \frac{|x-y|^2}{t}} \leq p_t^R(x,y) \leq C_2 t^{-d/2} e^{-c_2 \frac{|x-y|^2}{t}} \tag{11}$$

for constants $c_1, c_2, C_1, C_2$ and small enough $t$. This verifies the conditions of (Hsu, 1995, Theorem 1.2), yielding the conclusion. ☐

We now prove a general lemma on when the probability of hitting $B(x_j, s)$ before a time $t^*$ is asymptotically equal for two processes.

**Lemma S6.10.** *Let $Q$ be a diffusion process with transition densities $p_t^Q(x, y)$, and let $p_t^K(x, y)$ of $Q$ killed at some boundary. If for some $\lambda > 0$ and small enough $s$, we have for all $t < t^*$ and $x, y \in B$ that*

$$1 \geq \frac{p_t^K(x_i, x)}{p_t^Q(x_i, x)} \geq 1 - e^{-\lambda t^{-1}} \text{ and } 1 \geq \frac{p_t^K(x, y)}{p_t^Q(x, y)} \geq 1 - e^{-\lambda t^{-1}},$$

*then the probabilities $h^K(t^*)$ and $h^Q(t^*)$ that $K$ and $Q$ hit $B(x_j, s)$ before $t^*$ are asymptotically equal.*

*Proof.* Let $B = B(x_j, s)$, and consider $s$ small enough so that $B(x_j, s) \subset D$. For $x \in B$ and $t > 0$, let $h^K(t, x)$ and $h^Q(t, x)$ be the densities of the first passage time to $B$, and let $h^K(T)$ and $h^Q(T)$ be the probabilities that the respective first passage times are at most $T$. Note that $h^K(t, x) \leq h^Q(t, x)$. For $\star \in \{K, Q\}$, we have

$$h^\star(t, x) = p_t^\star(x_i, x) - \int_0^t \int_{y \in B} p_{t-\tau}^\star(y, x) h^\star(\tau, y) dy d\tau$$

so we may integrate to obtain

$$h^\star(T) = \int_0^T \int_{x \in B} p_t^\star(x_i, x) dt dx - \int_0^T \int_0^t \int_{x, y \in B} p_{t-\tau}^\star(y, x) h^\star(\tau, y) \, dy dx d\tau dt. \tag{12}$$

Define the differences $d(T) := h^Q(T) - h^K(T)$, $d(t, x) = h^Q(t, x) - h^K(t, x)$, and $e_t(x, y) := p_t^Q(x, y) - p_t^K(x, y)$. By assumption, if $x = x_i$ or $x, y \in B$, we have

$$e_t(x, y) \leq e^{-\lambda t^{-1}} p_t^Q(x, y).$$

Subtracting (12) for $\star \in \{K, Q\}$, we obtain

$$d(T) = \int_0^T \int_{x \in B} e_t(x_i, x) dt dx + \int_0^T \int_0^t \int_{x, y \in B} e_{t-\tau}(y, x) h^K(\tau, y) \, dy dx d\tau dt$$

$$+ \int_0^T \int_0^t \int_{x, y \in B} p_{t-\tau}^K(y, x) d(\tau, y) \, dy dx d\tau dt$$

$$\leq \int_0^T \int_{x \in B} e^{-\lambda t^{-1}} p_t^Q(x_i, x) dt dx + \int_0^T \int_0^t \int_{x, y \in B} e^{-\lambda(t-\tau)^{-1}} p_{t-\tau}^Q(y, x) h^K(\tau, y) \, dy dx d\tau dt$$

$$+ \int_0^T \int_0^t \int_{y \in B} d(\tau, y) d\tau dt dy$$

$$\leq \int_0^T e^{-\lambda t^{-1}} dt + \int_0^T \int_0^t \int_{y \in B} e^{-\lambda(t-\tau)^{-1}} h^K(\tau, y) \, dy d\tau dt + \int_0^T d(\tau) d\tau$$

$$\leq 2 \int_0^T e^{-\lambda t^{-1}} dt + \int_0^T d(\tau) d\tau$$

$$\leq 2T e^{-\lambda T^{-1}} + \int_0^T d(\tau) d\tau.$$

By Gronwall's inequality, this implies that

$$d(T) \leq 2T e^{-\lambda T^{-1}} + 2 \int_0^T \tau e^{-\lambda \tau^{-1}} (T - \tau) d\tau \leq 2(T + T^3) e^{-\lambda T^{-1}}.$$

We conclude that

$$\lim_{s \to \infty} d(t^*) = \lim_{s \to \infty} h^Q(t^*) - h^K(t^*) = 0. \qquad \square$$

**Lemma S6.11.** *As $s \to 0$, we have $\mathbb{P}(E_2 \mid E_1) \to 1$.*

*Proof.* Let $B = B(x_j, s)$. By Lemma 6.7 with $\rho'$ small enough, we have for some $\lambda > 0$ and small enough $s$ that for all $t < t^\star$ and $x, y \in B$ that

$$\frac{p_t^G(x_i, x)}{p_t^F(x_i, x)} \geq 1 - e^{-\lambda t^{-1}} \text{ and } \frac{p_t^G(x, y)}{p_t^F(x, y)} \geq 1 - e^{-\lambda t^{-1}}.$$

Notice that $\mathbb{P}(E_2)$ is the probability that the Brownian motion killed at $G(\rho)$ hits $B$ before $t^\star$ and $\mathbb{P}(E_1)$ is the probability that the free Brownian motion hits $B$ before $t^\star$. Therefore, Lemma 6.10 implies that

$$\lim_{s \to 0} \mathbb{P}(E_1) = \lim_{s \to 0} \mathbb{P}(E_2),$$

from which we conclude that

$$\lim_{s \to 0} \mathbb{P}(E_2 \mid E_1) = \lim_{s \to 0} \frac{\mathbb{P}(E_2)}{\mathbb{P}(E_1)} = 1. \qquad \square$$

**Lemma S6.12.** *As $s \to 0$, we have $\mathbb{P}(E_4 \mid E_3) \to 1$.*

*Proof.* Applying Lemma 6.10 twice using Lemmas 6.9 and 6.8 implies that

$$\lim_{s \to \infty} \mathbb{P}(E_1) = \lim_{s \to \infty} h^F(t^\star) = \lim_{s \to \infty} h^K(t^\star) = \lim_{s \to \infty} h^R(t^\star) = \lim_{s \to \infty} \mathbb{P}(E_3).$$

We conclude from Lemma 6.11 that

$$\lim_{s \to \infty} \mathbb{P}(E_4 \mid E_3) = \lim_{s \to \infty} \frac{\mathbb{P}(E_4)}{\mathbb{P}(E_3)} = \lim_{s \to \infty} \frac{\mathbb{P}(E_2)}{\mathbb{P}(E_1)} = \lim_{s \to \infty} \mathbb{P}(E_2 \mid E_1) = 1. \qquad \square$$

*Proof of Theorem 6.6.* Throughout this proof, we will take $t^\star = s^\gamma$ and $\widehat{\beta} = s^\alpha$ for some fixed $\alpha < 0$ and $\gamma > 0$ so that $\alpha + \gamma > 0$. We will pick a small $s > 0$ at the end of the proof.
**Bounding the effect of conditioning on $E_4$ on the process with boundary:** By Corollary 6.2, for any $\widehat{\beta}$ we have that

$$-\log(\mathbb{E}[\exp(-\widehat{\beta} g_n^2 \widehat{T}_{B(x_j,s),n}^{x_i})]) / \sqrt{2\widehat{\beta}} \xrightarrow{d} -\log(\mathbb{E}[\exp(-\widehat{\beta} \overline{T}_{B(x_j,s)}^{x_i})]) / \sqrt{2\widehat{\beta}}.$$

Conditioning on $E_3$ and $E_4$, we see that

$$\begin{aligned}
\mathbb{E}[\exp(-\widehat{\beta} \overline{T}_{B(x_j,s)}^{x_i})] &= \mathbb{E}[\exp(-\widehat{\beta} \overline{T}_{B(x_j,s)}^{x_i}) \mid E_3^c] \, \mathbb{P}(E_3^c) \\
&+ \mathbb{E}[\exp(-\widehat{\beta} \overline{T}_{B(x_j,s)}^{x_i}) \mid E_4] \, \mathbb{P}(E_4) \\
&+ \mathbb{E}[\exp(-\widehat{\beta} \overline{T}_{B(x_j,s)}^{x_i}) \mid E_3 \cap E_4^c] \, \mathbb{P}(E_4^c \mid E_3) \, \mathbb{P}(E_3).
\end{aligned}$$

By definition of $E_3$, we have $0 \leq \mathbb{E}[\exp(-\widehat{\beta} \overline{T}_{B(x_j,s)}^{x_i}) \mid E_3^c] \, \mathbb{P}(E_3^c) \leq e^{-\widehat{\beta} t^\star}$. By the trivial bound $\exp(-\widehat{\beta} \overline{T}_{B(x_j,s)}^{x_i}) \leq 1$, we find that

$$0 \leq \mathbb{E}[\exp(-\widehat{\beta} \overline{T}_{B(x_j,s)}^{x_i}) \mid E_3 \cap E_4^c] \, \mathbb{P}(E_4^c \mid E_3) \, \mathbb{P}(E_3) \leq 1 - \mathbb{P}(E_4 \mid E_3).$$

By Lemma 6.12, for any $\tau > 0$, for small enough $s > 0$ we have $e^{-\widehat{\beta} t^\star} < \tau$ and $1 - \mathbb{P}(E_4 \mid E_3) < \tau$. Noting also that $\mathbb{P}(E_4) = \mathbb{P}(E_2)$ and $\mathbb{E}[\exp(-\widehat{\beta} \overline{T}_{B(x_j,s)}^{x_i}) \mid E_4] = \mathbb{E}[\exp(-\widehat{\beta} \widehat{T}_{B(x_j,s)}^{x_i}) \mid E_2]$, we conclude for small enough $s$ that

$$\left| \mathbb{E}[\exp(-\widehat{\beta} \overline{T}_{B(x_j,s)}^{x_i})] - \mathbb{E}[\exp(-\widehat{\beta} \widehat{T}_{B(x_j,s)}^{x_i}) \mid E_2] \, \mathbb{P}(E_2) \right| < 2\tau. \tag{13}$$

**Bounding the effect of conditioning on $E_2$ on the process without boundary:** We now compare to the computations for Brownian motion without boundary. By conditioning on $E_1$ and $E_2$, we have that

$$\begin{aligned}
\mathbb{E}[\exp(-\widehat{\beta} \widehat{T}_{B(x_j,s)}^{x_i})] &= \mathbb{E}[\exp(-\widehat{\beta} \widehat{T}_{B(x_j,s)}^{x_i}) \mid E_1^c] \, \mathbb{P}(E_1^c) \\
&+ \mathbb{E}[\exp(-\widehat{\beta} \widehat{T}_{B(x_j,s)}^{x_i}) \mid E_2] \, \mathbb{P}(E_2) \\
&+ \mathbb{E}[\exp(-\widehat{\beta} \widehat{T}_{B(x_j,s)}^{x_i}) \mid E_1 \cap E_2^c] \, (1 - \mathbb{P}(E_2 \mid E_1)) \, \mathbb{P}(E_1).
\end{aligned}$$

We again note that

$$0 \leq \mathbb{E}[\exp(-\widehat{\beta}\widehat{T}^{x_i}_{B(x_j,s)}) \mid E_1^c] \, \mathbb{P}(E_1^c) \leq e^{-\widehat{\beta}t^\star}$$

and

$$0 \leq \mathbb{E}[\exp(-\widehat{\beta}\widehat{T}^{x_i}_{B(x_j,s)}) \mid E_1 \cap E_2^c] \, (1 - \mathbb{P}(E_2 \mid E_1)) \, \mathbb{P}(E_1) \leq 1 - \mathbb{P}(E_2 \mid E_1).$$

These together with Lemma 6.11 imply that for any $\tau > 0$, for small enough $s > 0$ we have that $e^{-\widehat{\beta}t^\star} < \tau$ and $1 - \mathbb{P}(E_2 \mid E_1) < \tau$. We conclude for small enough $s$ that

$$\left| \mathbb{E}[\exp(-\widehat{\beta}\widehat{T}^{x_i}_{B(x_j,s)})] - \mathbb{E}[\exp(-\widehat{\beta}\widehat{T}^{x_i}_{B(x_j,s)}) \mid E_2] \, \mathbb{P}(E_2) \right| < 2\tau. \tag{14}$$

Combining (13) and (14), we conclude for small enough $s$ that

$$\left| \mathbb{E}[\exp(-\widehat{\beta}\overline{T}^{x_i}_{B(x_j,s)})] - \mathbb{E}[\exp(-\widehat{\beta}\widehat{T}^{x_i}_{B(x_j,s)})] \right| < 4\tau. \tag{15}$$

**Aggregating the estimates:** To conclude, for any $\delta > 0$ and $\widehat{\beta}_0 > 0$, choose $\tau > 0$ small enough so that if $|x - y| < 4\tau$, then for all $\widehat{\beta} > \widehat{\beta}_0$, we have

$$\left| \log(x)/\sqrt{2\widehat{\beta}} - \log(y)/\sqrt{2\widehat{\beta}} \right| < \delta/3.$$

Now, choose $s > 0$ small enough and $n$ large enough so that $\widehat{\beta} > \widehat{\beta}_0$, and for this $\tau$, we have:

- by our previous discussion, (15) holds;
- by Lemma S6.1, we have

$$\left| - \log(\mathbb{E}[\exp(-\widehat{\beta}\widehat{T}^{x_i}_{B(x_j,s)})])/\sqrt{2\widehat{\beta}} - |x_i - x_j| \right| < \delta/3;$$

- by Corollary S6.2, we have

$$\left| - \log(\mathbb{E}[\exp(-\widehat{\beta}\overline{T}^{x_i}_{B(x_j,s)})])/\sqrt{2\widehat{\beta}} + \log(\mathbb{E}[\exp(-\widehat{\beta}g_n^2\widehat{T}^{x_i}_{B(x_j,s),n})])/\sqrt{2\widehat{\beta}} \right| < \delta/3.$$

For these choices of $\tau$, $s$, and $n$, we have by (15) that

$$\left| \log(\mathbb{E}[\exp(-\widehat{\beta}\overline{T}^{x_i}_{B(x_j,s)})])/\sqrt{2\widehat{\beta}} - \log(\mathbb{E}[\exp(-\widehat{\beta}\widehat{T}^{x_i}_{B(x_j,s)})])/\sqrt{2\widehat{\beta}} \right| < \delta/3.$$

Combining the last three inequalities yields the desired

$$\left| \log(\mathbb{E}[\exp(-\widehat{\beta}g_n^2\overline{T}^{x_i}_{B(x_j,s),n})])/\sqrt{2\widehat{\beta}} - |x_i - x_j| \right| < \delta. \qquad \square$$

# 7  1-D bias calculation

We repeat the full theorem statement and proof for the bias characterization.

**Theorem S7.1.** *Let $T^{x_i}_{x_j}$ be the hitting time to $x_j$ of a 1-dimensional Itô process with drift $\mu(x) = \frac{\partial \log(p(x))}{\partial x}\overline{\varepsilon}^2(x)$ and diffusion $\overline{\varepsilon}^2(x)$ started at $x_i$ with reflecting boundary $\gamma$ for $\gamma < x_i < x_j$. The Laplace transform of $T^{x_i}_{x_j}$ admits the asymptotic expansion*

$$\mathbb{E}[-\exp(\beta T^{x_i}_{x_j})] = \frac{c_1}{f(x_i)^{1/4}p(x_i)} \exp\left( -\sqrt{\beta} \int_{x_i}^{x_j} \sqrt{f(s)}ds \right)$$
$$\left( 1 + \left( 1 + o\left( \frac{1}{\sqrt{\beta}} \right) \right) \exp\left( -2\sqrt{\beta} \int_{\gamma}^{x_i} \sqrt{f(x)}dx \right) + o(\exp(-\beta)) \right),$$

*where $f(x) = \frac{2}{\overline{\varepsilon}(x)^2} + \frac{1}{\beta}\frac{\partial \log(p(x))}{\partial x^2} + \frac{1}{\beta}\left( \frac{\partial \log(p(x))}{\partial x} \right)^2$, and $c_1$ is a normalization constant depending on $p, \overline{\varepsilon}$, and $j$ to make $E[-\beta T^{x_i}_{x_j}] = 1$.*

*Proof.* Let $\mathbb{E}[\exp(-\beta T_{x_j}^{x_i})] = u(x_i)$, where $u(x)$ is the hitting time to $x_j$ from point $x$. By Feynman-Kac, this is

$$\frac{\partial^2 u}{\partial x^2} + 2\frac{\partial \log(p(x))}{\partial x}\frac{\partial u}{\partial x} + q(x)u = 0,$$

where $q(x) = -2\beta\bar{\varepsilon}(x)^{-2}$. Rewrite this as a perturbation of a second order ODE via the change of variables to obtain

$$y(x) = u(x)\exp\left(\int_\gamma^x \frac{\partial \log(p(y))}{\partial y}dy\right) = u(x)p(x)p(\gamma)^{-1}$$

$$f(x) = \frac{2}{\bar{\varepsilon}(x)^2} + \frac{1}{\beta}\left(\frac{\partial \log(p(x))}{\partial x^2} + \left(\frac{\partial \log(p(x))}{\partial x}\right)^2\right)$$

$$\frac{1}{\beta}\frac{\partial^2 y}{\partial x} = f(x)y(x).$$

Since this is a type of Schrodinger's equation with $f(x) \neq 0$ everywhere we can apply the WKBJ asymptotic expansion (Bender & Orszag, 1999, section 10.1) to obtain

$$y(x) = \frac{c_1}{f(x)^{1/4}}\exp\left(-\sqrt{\beta}\int_{x_0}^x \sqrt{f(s)}ds\right) + \frac{c_2}{f(x)^{1/4}}\exp\left(\sqrt{\beta}\int_{x_0}^x \sqrt{f(s)}ds\right) + o(\exp(-\beta)).$$

Since we assumed $x_i < x_j$ and by the boundary condition $u(x_j) = 1$ we have

$$u(x) = \frac{c_2 p(\gamma)}{f(x)^{1/4}p(x)}\exp\left(-\sqrt{\beta}\int_x^{x_j} \sqrt{f(s)}ds\right) + \frac{c_1 p(\gamma)}{f(x)^{1/4}p(x)}\exp\left(\sqrt{\beta}\int_x^{x_j} \sqrt{f(s)}ds\right) + o(\exp(-\beta)).$$

To obtain the boundary conditions, note that $u'(\gamma) = 0$. Taking the derivative for $y(x)p(x)$, setting to zero and solving for $c_2$ results in

$$c_2 = c_1 \frac{\exp(-2\sqrt{\beta}\int_\gamma^{x_j}\sqrt{f(s)}ds)(p(\gamma)4\sqrt{\beta}f(\gamma)^{3/2} + f'(\gamma)) - f(\gamma)p'(\gamma)}{4\sqrt{\beta}f(\gamma)^{3/2}p(\gamma) - p(\gamma)f'(\gamma) + 4f(\gamma)p'(\gamma)} + o(\exp(-\beta)),$$

from which we obtain

$$c_2 = c_1\exp\left(-2\sqrt{\beta}\int_\gamma^{x_j}\sqrt{f(s)}ds\left(1 + o\left(\sqrt{\frac{1}{\beta}}\right)\right)\right).$$

Pulling out the $-\sqrt{\beta}$ term, we get

$$u(x_i) = \mathbb{E}[\exp(-\beta T_{x_j}^{x_i})] = \frac{c_1 p(\gamma)}{f(x_i)^{1/4}p(x_i)}\exp\left(-\sqrt{\beta}\int_{x_i}^{x_j}\sqrt{f(s)}ds\right)$$

$$\left(1 + \left(1 + o\left(\frac{1}{\sqrt{\beta}}\right)\right)\exp\left(-2\sqrt{\beta}\int_\gamma^{x_i}\sqrt{f(x)}dx\right) + o(\exp(-\beta))\right). \quad \square$$

We now connect this statement to the discrete walk.

**Corollary S7.2.** *Let $T_{B(x_j,s),n}^{x_i}$ be the discrete hitting time to a $s$ ball around $x_j$ where $s$ is selected as given in Theorem S2.13. Then the simple random walk over a graph constructed on density $p(x)$ and scale $\bar{\varepsilon}(x)$ has the following log-LTHT under the boundary conditions of Theorem S4.6*

$$-\log(\mathbb{E}[\exp(-\beta T_{B(x_j,s),n}^{x_i}g_n^2)])/\sqrt{2\beta} \to \int_{x_i}^{x_j}\sqrt{\frac{1}{\bar{\varepsilon}(x)} + \frac{1}{\beta}\left(\frac{\partial \log(p(x))}{\partial x^2} + \left(\frac{\partial \log(p(x))}{\partial x}\right)^2\right)}dx$$

$$+ \frac{\log(p(x_i)/p(x_j)) + \log(f(x_i)/f(x_j))/4}{\sqrt{2\beta}} + o(\log(1 + e^{-\sqrt{2\beta}})/\sqrt{2\beta}).$$

*Proof.* Taking the logarithm of the result of Theorem 7.1 and noting the initial condition $u(x_j) = 1$ implies that asymptotically we have

$$c_1 \propto \left( \frac{1}{f(x_j)^{1/4} p(x_j)} (1 + o(e^{-2\sqrt{\beta}})) \right)^{-1} \to f(x_j)^{1/4} p(x_j),$$

which completes the continuous statement. The convergence of the hitting time to its discrete counterpart follows from Theorem S2.13. □

## 8 Basic noise resistance

We give details for the basic noise bound from the main text footnote. Our goal is to prove the following statement about random walks.

**Theorem S8.1.** *Let $G_n$ be generated by the noise model of definition 4.7 with $\sum_j q_j = o(g_n^2)$. Then the simple random walk over $G_n$ converges to the same limit as the noiseless case in Theorem 2.2.*

*Proof.* Since the boundaries of both noisy and noiseless graphs are identical, we need only verify the moment conditions in the proof of Theorem 2.2. In particular we require that under any noise $q$, we have

$$\lim_{n \to \infty} g_n^{-2} \mathbb{E}[X_{t+1}^n - X_t^n | X_t^n] = \nabla \log(p(X_t^n)) \bar{\varepsilon}(X_t^n)^2$$

$$\lim_{n \to \infty} g_n^{-2} \mathsf{Cov}[X_{t+1}^n | X_t^n] = \bar{\varepsilon}(X_t^n)^2 \cdot I_n$$

$$\lim_{n \to \infty} g_n^{-2} \mathbb{E}[|X_{t+1}^n - X_t^n|^{2+\alpha} \mid X_t^n] = 0,$$

which we show in the Lemma S8.2 and S8.3 below. By the Stroock-Varadhan criterion, this implies convergence to Theorem 2.2, as well as any macroscopic quantities such as hitting times, or LTHTs with $\beta = \Theta(g_n^2)$. □

We now prove the moment bounds required for convergence of the noisy graph.

**Lemma S8.2** (Noisy moments). *If the noisy graph $G_n$ is generated by the noise model of Definition 4.7, for any choice of latent noise parameters $q_j$ such that $\sum_j q_j = o(g_n^2)$ then we have for $\alpha > 0$ that*

$$\lim_{n \to \infty} g_n^{-2} \mathbb{E}[X_{t+1}^n - X_t^n | X_t^n] = \nabla \log(p(X_t^n)) \bar{\varepsilon}(X_t^n)^2$$

$$\lim_{n \to \infty} g_n^{-2} \mathsf{Cov}[X_{t+1}^n | X_t^n] = \bar{\varepsilon}(X_t^n)^2 \cdot I_n$$

$$\lim_{n \to \infty} g_n^{-2} \mathbb{E}[|X_{t+1}^n - X_t^n|^{2+\alpha} \mid X_t^n] = 0.$$

*Proof.* Let $\overline{X}$ denote quantities in the noise-free graph. We recall from (Hashimoto et al., 2015, Theorem 3.3) that

$$\lim_{n \to \infty} g_n^{-2} \mathbb{E}[\overline{X}_{t+1}^n - \overline{X}_t^n | \overline{X}_t^n = x] = \nabla \log(p(x)) \bar{\varepsilon}(x)^2$$

$$\lim_{n \to \infty} g_n^{-2} \mathsf{Cov}[\overline{X}_{t+1}^n | \overline{X}_t^n = x] = \bar{\varepsilon}(x)^2 \cdot I_n$$

$$\lim_{n \to \infty} g_n^{-2} \mathbb{E}[|\overline{X}_{t+1}^n - \overline{X}_t^n|^{2+\alpha} \mid \overline{X}_t^n = x] = 0.$$

Let $\widehat{q} = \sum_i q_i$ so that $\widehat{q} = o(g_n^2)$. In the noisy graph, we first check the expectation via

$$\lim_{n \to \infty} g_n^{-2} \mathbb{E}[X_{t+1}^n - X_t^n \mid X_t^n = x] = \lim_{n \to \infty} (1 - \widehat{q}) g_n^{-2} \mathbb{E}[\overline{X}_{t+1}^n - \overline{X}_t^n \mid \overline{X}_t^n = x] + g_n^{-2} \sum_i q_i (x_i - x)$$

$$= \lim_{n \to \infty} g_n^{-2} \mathbb{E}[\overline{X}_{t+1}^n - \overline{X}_t^n | \overline{X}_t^n = x]$$

$$= \nabla \log(p(x)) \bar{\varepsilon}(x)^2.$$

The covariance follows because for all indices $i$ and $j$ we have

$$\lim_{n\to\infty} g_n^{-2}\mathbb{E}[(X_{t+1}^n - X_t^n)_i(X_{t+1}^n - X_t^n)_j \mid X_t^n = x]$$

$$= \lim_{n\to\infty}(1 - \widehat{q})g_n^{-2}\mathbb{E}[(X_{t+1}^n - X_t^n)_i(X_{t+1}^n - X_t^n)_j \mid \overline{X}_t^n = x] + g_n^{-2}\sum_k q_k(x_k - x)_i(x_k - x)_j$$

$$= \lim_{n\to\infty} g_n^{-2}\mathbb{E}[(\overline{X}_{t+1}^n - \overline{X}_t^n)_i(\overline{X}_{t+1}^n - \overline{X}_t^n)_j \mid \overline{X}_t^n = x]$$

$$= \delta_{ij}\overline{\varepsilon}(x)^2.$$

Finally, the higher moments follow because we have

$$\lim_{n\to\infty} g_n^{-2}\mathbb{E}[|X_{t+1}^n - X_t^n|^{2+\alpha} \mid X_t^n = x]$$

$$= \lim_{n\to\infty} g_n^{-2}(1 - \widehat{q})g_n^{-2}\mathbb{E}[|\overline{X}_{t+1}^n - \overline{X}_t^n|^{2+\alpha} \mid \overline{X}_t^n = x] + g_n^{-2}\sum_i q_i|x_i - x|^{2+\alpha}$$

$$= \lim_{n\to\infty} g_n^{-2}(1 - \widehat{q})g_n^{-2}\mathbb{E}[|\overline{X}_{t+1}^n - \overline{X}_t^n|^{2+\alpha} \mid \overline{X}_t^n = x]$$

$$= 0,$$

where we use that $|x_i - x|^{2+\alpha} = O(1)$. $\qquad\square$

**Lemma S8.3** (Strong LLN for noisy moments). *For a function $f(x)$ such that $\sup_{x\in B(0,\varepsilon)}|f(x)| < \varepsilon$ and $\sup_{x\in D}|f(x)| < C$ for some constant $C$, given $(\star)$ we have uniformly in $x \in \mathcal{X}_n$ that*

$$g_n^{-2}\sum_{y\in\mathsf{NB}_n(x)}\frac{1}{|\mathsf{NB}_n(x)|}f(y - x) \overset{a.s.}{\to} g_n^{-2}\int_{y\in B(x,\varepsilon_n(x))}f(y - x)\frac{p(y)}{p_{\varepsilon_n(x)}(x)}dy.$$

*Proof.* Denote the claimed value of the limit by $\mu(x)$. Let the set of non-noise out-neighbors of $x$ be $\overline{\mathsf{NB}}_n(x)$ and the set of noise out-neighbors of $x$ be $\widetilde{\mathsf{NB}}_n(x)$, where we consider noise edges to be strictly non-geometric edges. We have uniformly in $x \in cX$ that

$$g_n^{-2}\frac{\sum_{y\in\overline{\mathsf{NB}}_n(x)}f(y - x) + \sum_{y\in\widetilde{\mathsf{NB}}_n(x)}f(y - x)}{|\overline{\mathsf{NB}}_n(x)| + |\widetilde{\mathsf{NB}}_n(x)|} = g_n^{-2}\frac{\sum_{y\in\overline{\mathsf{NB}}_n(x)}f(y - x) + o(Cg_n^2)}{|\overline{\mathsf{NB}}_n(x)| + o(g_n^2)}$$

$$\overset{a.s.}{\to} g_n^{-2}\sum_{y\in\overline{\mathsf{NB}}_n(x)}\frac{f(y - x)}{|\overline{\mathsf{NB}}_n(x)|},$$

so the result follows by the noise-less result in (Hashimoto et al., 2015). $\qquad\square$

Now the behavior of noisy hitting times can be recovered by combining Lemma S8.1 with the convergence result of Corollary S6.2.

**Theorem S8.4.** *Let $G_n$ be a noisy geometric graph with noise $\sum_j q_j = o(g_n^2)$. For any $\delta$, there exists some $\beta = \widehat{\beta}g_n^2$, $s$, $c$ such that*

$$\left| -\frac{\log(\mathbb{E}[\exp(-\beta T_{B(x_j,s),n}^{x_i})])}{\sqrt{2\beta}}g_n - c|x_i - x_j| \right| \le \delta$$

*with high probability as $n \to \infty$.*

*Proof.* By Lemma S8.1, the noisy and noise-free walks converge to the same continuum limit, and this guarantees that by Corollary S6.2 that their hitting times converge in distribution. Applying Theorem 4.5 gives the desired result. $\qquad\square$

This is a basic, but useful result for robustness of hitting times. Up to $o(1)$ noise edges can be allowed for each vertex without disrupting the global convergence of hitting times.

(a) Modified LTHT rapidly converges to RA index.

(b) Conditioning on $t > 1$ substantially outperforms naive LTHT.

# 9 Resource allocation index

Recall that the directed RA index was defined by

$$R_{ij} = \sum_{x_k \in \mathsf{NB}_n(x_i) \cap \mathsf{NB}_n^{\text{in}}(x_j)} \frac{1}{|\mathsf{NB}_n(x_k)|}$$

and the modified log-LTHT was defined by

$$M_{ij}^{\text{mod}} = -\log(\mathbb{E}[\exp(-\beta T_{x_j,n}^{x_i}) \mid T_{x_j,n}^{x_i} > 1]).$$

## 9.1 RA index reduction

**Theorem S9.1.** *If $\beta = \omega(\log(g_n^d n))$ and $x_i$ and $x_j$ have at least one common neighbor, then*

$$M_{ij}^{mod} - 2\beta \to -\log(R_{ij}) + \log(|\mathsf{NB}_n(x_i)|).$$

*Proof.* Let $P_{ij}(t)$ be the probability of going from $x_i$ to $x_j$ in $t$ steps, and $H_{ij}(t)$ the probability of not hitting before time $t$. Factoring the two-step hitting time yields

$$M_{ij}^{\text{mod}} = 2\beta - \log(P_{ij}(2)) - \log\left(1 + \sum_{t=3}^{\infty} \frac{P_{ij}(t)}{P_{ij}(2)} H_{ij}(t) e^{-\beta(t-2)}\right).$$

Let $k_{\max}$ be the maximal out-degree which occurs in $G_n$. By assumption, at least one of the at most $k_{\max}^2$ two-step paths from $x_i$ goes to $x_j$, we have the bound $\frac{P_{ij}(t)}{P_{ij}(2)} \le k_{\max}^2$. For $\beta = \omega(\log(g_n^d n))$, we see that $\beta = \omega(2\log(k_{\max}))$ with high probability. Applying the bounds $H_{ij}(t) \le 1$ and $\frac{P_{ij}(t)}{P_{ij}(2)} \le k_{\max}^2$, we obtain

$$\sum_{t=3}^{\infty} \frac{P_{ij}(t)}{P_{ij}(2)} H_{ij}(t) e^{-\beta(t-2)} \le \frac{k_{\max}^2}{e^\beta - 1} = o(k_{\max}^{-1}).$$

We conclude that $M_{ij}^{\text{mod}} \to 2\beta - \log(P_{ij}(2))$. It remains to verify that $\log(P_{ij}(2))$ is related to the resource allocation index by

$$\log(P_{ij}(2)) = \log\left(\frac{1}{|\mathsf{NB}_n(x_i)|} \sum_{k \in \mathsf{NB}_n(x_i) \cap \mathsf{NB}_n^{in}(x_j)} \frac{1}{\mathsf{NB}_n(x_k)}\right) = \log(R_{ij}) - \log(|\mathsf{NB}_n(x_i)|). \qquad \square$$

## 9.2 RA index robustness

We verify the robustness of the RA index by directly bounding the statistics involved.

**Theorem S9.2.** *If $q_i = q = o(g_n^{d/2})$ for all $i$, then for any $\delta > 0$ there exist cutoffs $c_1, c_2$ and scaling $h_n$ so that with probability at least $1 - \delta$, for any $i, j$ we have*

- $|x_i - x_j| < \min\{\varepsilon_n(x_i), \varepsilon_n(x_j)\}$ *if $R_{ij}h_n < c_1$;*

- $|x_i - x_j| > 2\max\{\varepsilon_n(x_i), \varepsilon_n(x_j)\}$ *if $R_{ij}h_n > c_2$.*

*Proof.* Decompose the out-degree of $x_i$ into expectation and noise terms by

$$|\mathsf{NB}_n(x_i)| = nq + k_i + z_i,$$

where $k_i = \varepsilon_n(x_i)^d p(x_i) V_d n$, $V_d$ is the volume of the $d$-unit ball, and $z_i$ is a random variable giving the remaining error. The number of noise edges has a binomial distribution with $n$ draws and success probability $q$, and the number of geometric edges has a Poisson distribution with rate $k_i$. Therefore, the Chebyshev inequality implies

$$\mathbb{P}(|z_i| > c) \leq \frac{k_i + nq(1-q)}{c^2} < \frac{k_i + nq}{c^2}. \tag{16}$$

Let $\delta_1 = \delta/4$ and define $c$ by the equality

$$\delta_1 = \frac{k_i + nq(1-q)}{c^2} \tag{17}$$

so that $c = \delta_1^{-1/2}\sqrt{k_i + nq}$. For the rest of the proof, we condition on the event that $|z_i| < c$. By Taylor expanding $\frac{1}{|\mathsf{NB}_n(x_i)|}$ in $z_i$, we have that

$$\frac{1}{|\mathsf{NB}_n(x_i)|} = \frac{1}{nq + k_i} - \frac{z_i}{(nq + k_i)^2} + O\left(\frac{z_i^2}{(nq + k_i)^3}\right)$$

$$= \frac{1}{nq + k_i} - O\left(\frac{c}{(nq + k_i)^2}\right). \tag{18}$$

By (17), we see that $|z_i| < c$ with probability at least $1 - \delta_1$, which implies that

$$\frac{c}{(nq + k_i)^2} < \delta_1^{-1/2}(nq + k_i)^{-3/2}.$$

By the definition of the RA index, we obtain

$$R_{ij} = \sum_{x_k \in \mathsf{NB}_n(x_i) \cap \mathsf{NB}_n^{\text{in}}(x_j)} \left(\frac{1}{nq + k_i} + O(\delta_1^{-1/2}(nq + k_i)^{-3/2})\right).$$

Since our domain is compact, we may define

$$k_n^+ = \sup_x \varepsilon_n(x)^d p(x) V_d n \qquad \text{and} \qquad k_n^- = \inf_x \varepsilon_n(x)^d p(x) V_d n.$$

By construction, $k_n^+ > k_i > k_n^-$ for all $i$. Let $C_{ij} := |\mathsf{NB}_n(x_i) \cap \mathsf{NB}_n^{\text{in}}(x_j)|$. Then we have

$$\frac{C_{ij}}{nq + k_n^+} - O\left(\frac{C_{ij}}{\delta_1^{1/2}(nq + k_n^+)^{3/2}}\right) \leq R_{ij} \leq \frac{C_{ij}}{nq + k_n^-} + O\left(\frac{C_{ij}}{\delta_1^{1/2}(nq + k_n^-)^{3/2}}\right). \tag{19}$$

Choose the scaling

$$h_n = \frac{nq + k_n^+}{k_n^+}.$$

We will now bound $C_{ij}$ to control $h_n R_{ij}$. To do this, decompose $C_{ij}$ as

$$C_{ij} = C_{ij}^g + C_{ij}^{n1} + C_{ij}^{n2},$$

where $C_{ij}^g$, $C_{ij}^{n1}$, and $C_{ij}^{n2}$ are defined as follows.

1. Geometric edges $(C^g_{ij})$: If $|x_i - x_j| < \min\{\varepsilon_n(x_i), \varepsilon_n(x_j)\}$ then they share common neighbors due to the geometric graph. Specifically their number of common neighbors has Poisson distribution with mean at least $\tau_d k_i(1-q)$, where $\tau_d$ is a constant independent of $n$ defined as the overlapping density of two kernels at a unit distance.

2. One noise edge $(C^{n1}_{ij})$: The edge $x_i \to x_k$ occurs by noise but $x_k \to x_j$ is geometric. There are at most $k^+_n$ such vertices with in-edges to $x_j$ and so this is at most a binomial random variable with $k^+_n$ draws and success probability $q$.

3. Two noise edges $(C^{n2}_{ij})$: Both $x_i \to x_k$ and $x_k \to x_j$ may occur by noise, this is at most a binomial random variable with $n - k^-_n$ draws and success probability $q^2$.

**The case of** $|x_i - x_j| < \min\{\varepsilon_n(x_i), \varepsilon_n(x_j)\}$**:** All types of edges may occur, so we obtain the moment bounds

$$\mathbb{E}\left[\frac{C_{ij}}{k^+_n}\right] \geq \tau_d(1-q)\frac{k^-_n}{k^+_n}$$

$$\mathsf{Var}\left[\frac{C_{ij}}{k^+_n}\right] \leq \frac{\tau_d(1-q)k^-_n}{(k^+_n)^2} + \frac{(n-k^-_n)q^2(1-q^2) + k^+_n q(1-q)}{(k^+_n)^2} < \frac{\tau_d(1-q)k^-_n}{(k^+_n)^2} + \frac{nq^2 + k^+_n q}{(k^+_n)^2}.$$

Notice that $\frac{k^+_n}{k^-_n}$ is bounded between the minimum and maximum of $\frac{\varepsilon_n(x)p(x)}{\varepsilon_n(y)p(y)}$ for $x, y \in D$, so $\lim_{n\to\infty} \mathbb{E}\left[\frac{C_{ij}}{k^+_n}\right] \geq c_{ij}$ for some $c_{ij} > 0$. Further, we find that $\mathsf{Var}\left[\frac{C_{ij}}{k^+_n}\right] \to 0$. These imply that for large enough $n$, we have $\frac{C_{ij}}{k^+_n} > c_{ij}$ with probability at least $1 - \delta_1$. Therefore, if $|x_i - x_j| < \min\{\varepsilon_n(x_i), \varepsilon_n(x_j)\}$, we have

$$\lim_{n\to\infty} h_n R_{ij} \geq \lim_{n\to\infty} \frac{C_{ij}}{k^+_n} - O\left(\frac{C_{ij}}{\delta^{1/2}(nq + k^+_n)^{1/2}k^+_n}\right) \geq c_{ij}. \tag{20}$$

**The case of** $|x_i - x_j| > 2\max\{\varepsilon_n(x_i), \varepsilon_n(x_j)\}$**:** Only noise cases occur, hence we have the moment bounds

$$\mathbb{E}\left[\frac{C_{ij}}{k^-_n}\right] \leq \frac{q^2(n-k^-_n) + qk^+_n}{k^-_n} < \frac{q^2 n}{k^-_n} + \frac{qk^+_n}{k^-_n}$$

$$\mathsf{Var}\left[\frac{C_{ij}}{k^-_n}\right] \leq \frac{(n-k^-_n)q^2(1-q^2) + k^+_n q(1-q)}{(k^-_n)^2} < \frac{qn}{(k^-_n)^2} + \frac{qk^+_n}{(k^-_n)^2}.$$

Because $q = o(g_n^{d/2})$, both the expectation and variance converge to zero and for large enough $n$, we have $C_{ij}/k^-_n \to 0$ probability at least $1 - \delta_1$. Therefore, if $|x_i - x_j| > 2\max\{\varepsilon_n(x_i), \varepsilon_n(x_j)\}$, for we have

$$\lim_{n\to\infty} h_n R_{ij} \leq \lim_{n\to\infty} \frac{C_{ij}(nq + k^+_n)}{k^+_n(nq + k^-_n)} + O\left(\frac{C_{ij}(nq + k^+_n)}{\delta^{1/2}k^+_n(nq + k^-_n)^{3/2}}\right) \to 0. \tag{21}$$

**Combining the cases:** Taking $h_n = \frac{nq + k^+_n}{k^+_n}$, we combine (20) and (21) to conclude that the desired holds with probability at least $1 - 3\delta_1 > 1 - \delta$ for any $c_1 \leq c_{ij}$ and $c_2 > 0$. $\qquad\square$

## Footnotes

[1] Note that there is a typographical error in Hashimoto et al. (2015) adding an additional factor of $\varepsilon_n(x)^{-d}$.

[2]Note that there is a typographical error in Hashimoto et al. (2015) adding an additional factor of $\varepsilon_n(x)^{-d}$.