[Reviews · NeurIPS 2015]

Submitted by Assigned_Reviewer_1

Abstract: Large unweighted directed graphs are commonly used to capture relations be- tween entities. A fundamental problem in the analysis of such networks is to properly define the similarity or dissimilarity between any two vertices. Despite the significance of this problem, statistical characterization of the proposed met- rics has been limited.

What particular problem do the authors have in mind? The same argument is true for weighted, undirected graphs but they seem to be excluded. If one would know what problem the authors are talking about this would become clear. In the paper I could not find any connection to some 'real world' problem but only a mathematical consideration of the problem.

How about betweenness centrality, as used to identify bottleneck genes in gene networks. http://journals.plos.org/ploscompbiol/article?id=10.1371/journal.pcbi.0030059 Is that a problem the author has in mind? Can one compare both methods? Compare with any other method?

Without a clear connection to a real world problem I consider the paper out of scope for NIPS. However, I think the paper could make a nice contribution for a mathematics conference/journal.

The paper concludes without a summary/discussion.

Summary: The paper presents a mathematical analysis of random walks on networks.

Submitted by Assigned_Reviewer_2

The paper rigorously evaluates the consistency, cluster-preservation and robustness of an improved hitting-time metric to measure the similarity between any two vertices of a large unweighted directed graph.

This paper's main contribution is a new set of techniques for analyzing random walks on graphs using stochastic calculus.

The theoretical proofs are mostly clear and convincing but the presentation of the empirical results must be improved.

The paper also lacks a conclusion's section as it abruptly ends after presenting very briefly (half a page), the experimental setup, results, and discussion.

The experiments section should also be improved by including justification regarding:

+ The use of different LTHT configurations for each of the two experiments; + The choice for the largest vertex degree (5) to consider two separate links prediction and for the beta value (0.2)

Finally, the discussion should include more insights about the results, for example, the higher true positive rate of "One-step LTHT" when the false positive rate is very low (Figure 3).
Summary: This paper's main contribution is a new set of techniques for analyzing random walks on graphs using stochastic calculus. The theoretical proofs are mostly clear and convincing but the presentation of the empirical results must be largely improved.

Submitted by Assigned_Reviewer_3

To investigate the similarity or distance between any two vertices in graph, the authors develop a class of techniques for analyzing random walks on spatial graphs using stochastic calculus. For example, the authors give explanation for the problem of degeneracy of expected hitting times. Furthermore, they propose a metric based on the Laplace trans-formed hitting time (LTHT) and rigorously evaluate its consistency, cluster-preservation, and robustness under a general network model which encapsulates the latent space assumption.Besides the test on simulated data, the authors perform tests on two kinds of real-world data for link prediction and show that the LTHT matches theoretical predictions and outperforms the baseline methods like RA, shortest path length and common neighbors. Moreover, the author also briefly introduce two ways, namely matrix inversion and sampling,to algorithmically compute the LTHT similarity between vertex. Overall, it is a rigorously theoretical work on novel similarity approaches on graph with necessary comparative experiments.
Summary: This manuscript develops a class of techniques for analyzing random walks on spatial graphs using stochastic calculus and proposes a well defined metric(LTHT) to calculate the vertex similarity in spatial graph. It is a rigorously theoretical work on novel similarity approaches on graph with necessary comparative experiments.

Submitted by Assigned_Reviewer_4

This paper proposes a novel metric called Laplace transformed hitting time (LTHT) to measure the similarity between vertexes. Several properties of this metric are provided and proven. Experiments on both synthetic data and real-world data have demonstrated its superiority to other baseline measures. The structure of the paper is very difficult to follow. It is expected to start with the LTHT definition directly. But a spatial graph definition and some major results from [6] are provided first. It is unclear how generalizable of the conclusions in the paper given the underlying graph generative model is different.

The technical writing is also very dry. Notations within theorems are usually given with no explanations.

The experimental section is too short. Also, one-step LTHT and two-step LTHT are given in the experimental results without further explanations.
Summary: In all, this paper may contain some interesting discovery, but the writing is very difficult to follow, in terms of both structure and technical details.

Author Feedback
Author rebuttal: We thank the reviewers for their helpful comments and will include suggested improvements in the camera-ready version. We first address some general points made by multiple reviewers:

* Brevity of experimental description and lack of conclusion: These are due to the eight-page space constraint of the NIPS format. We chose to focus our presentation on providing exposition for the theoretical tools developed for hitting time metrics. In the supplement, we have provided additional simulated experiments that support our theoretical claims (figs 1-6) and have full details of the run in the ipython notebooks. We will address these concerns by including an extended edition of the text with full experimental details (some of which already appear in the supplemental iPython notebook) as well as extended discussion and conclusions.

* Technical focus of the paper: We chose to focus on the technical rather than experimental and applied aspects of the work because we believe that both the results and proof methods we introduce to understand hitting times will admit broader usage in developing statistically rigorous graph algorithms. Because the tools from stochastic processes we apply are less familiar to the community, we felt it was important to devote more space to giving a clear and complete exposition of our technique. While we provide a short experimental validation of the LTHT to validate our claims, we did not attempt a fully comprehensive comparison due to space constraints and the fact that prior comparisons of link prediction metrics already exist in the literature [8], [21].

Specific comments:
Reviewer_3
1. Structure of the paper: The definition of a spatial graph is a strict generalization of the model considered in [6]. The results shown in the framework of [6] apply equally well to the spatial graph model proposed in this paper with similar proofs. We gave full definitions and explanations here to provide the reader with context and intuition for our results on LTHT and to clearly state extensions to the results of [6] which we prove and use. We felt this was especially important because techniques based on convergence to stochastic processes have only recently appeared in the literature and use tools which are likely to be unfamiliar to many readers.
2. Notation:
3. Shortness of experimental section: The two-step LTHT is defined on line 368 of Section 4.3, and one-step LTHT is the same as LTHT. To clarify this point, we will use "LTHT" instead of "one-step LTHT" and add a pointer to the definition of the two-step LTHT in the results section.

Reviewer_4
1. Use of two different LTHT configurations: The citation network is unweighted, hence every link prediction query is between two vertices which do not share a link. This means that the one and two step log-LTHT's are identical in this experiment.

In the associative thesaurus data, the evaluation is weighted, and we consider whether we can recover high and low strength edges. Therefore, there is a choice of whether or not to remove direct links, and we can evaluate both one and two-step LTHT's. This point will be clarified in the final version of the text.
2. Largest vertex degree filtering: The vertex degree threshold is to improve the performance of the baseline RA-index and common neighbors algorithms. Both of these algorithms perform very badly due to ties without a degree threshold, and we found the low performance of these baseline algorithms to be unrepresentative.

The beta value was chosen to be the reciprocal of the vertex degree, as scaling by inverse degree is similar to the g_n^{-1} scaling described in the paper. This parameter value is not the result of hand-tuning, and beta values of 0.1 and 1 give qualitatively similar performance on both tasks (For beta = 0.1, 0.2, 1: figure 2 LTHT performance is 38,39,36. figure 3 LTHT AUC is 0.88, 0.89,0.87 respectively). We will include these additional details in the final supplement.
3. Further detail on LTHT results: Discussion of detailed results and beta parameter value selection will be included in an extended discussion section in the camera-ready version of the supplement.

Reviewer_5
1. What problem do we solve?: The concrete problem we have used to illustrate and validate our method is link prediction, as described in Lu and Zhou ([9])
2. Weighted graphs: The same algorithm can be applied in the weighted, directed case, but in this setting additional assumptions on the weight and latent distance are necessary in order to prevent the problem from being ill-posed.
3. Why not betweenness: Betweeness centrality measures the importance of a single vertex, while our algorithm recovers the similarity of two vertices. These two quantities are incomparable.
4. Comparisons: Our experimental section compares the log-LTHT against other link prediction methods following the procedure of [9].